# Advancing ensemble learning techniques for residential building electricity consumption forecasting: Insight from explainable artificial intelligence

**Jihoon Moon[1,2☯], Muazzam Maqsood[3☯], Dayeong So[2], Sung Wook Baik[4], Seungmin Rho[5]\*, Yunyoung Nam[2,6]\***

1 Department of AI and Big Data, Soonchunhyang University, Asan, Republic of Korea, 2 Department of ICT Convergence, Soonchunhyang University, Asan, Republic of Korea, 3 Department of Computer Science, COMSATS University Islamabad, Attock Campus, Attock, Pakistan, 4 Sejong University, Seoul, Republic of Korea, 5 Department of Industrial Security, Chung-Ang University, Seoul, Republic of Korea, 6 Department of Computer Science and Engineering, Soonchunhyang University, Asan, Republic of Korea

☯ These authors contributed equally to this work.
\* smrho@cau.ac.kr (SR); ynam@sch.ac.kr (YN)

**Data Availability Statement:** The manuscript includes comprehensive information about the dataset used. The "Appliances Energy Prediction Data Set," utilized in this research, is openly

## Abstract

Accurate electricity consumption forecasting in residential buildings has a direct impact on energy efficiency and cost management, making it a critical component of sustainable energy practices. Decision tree-based ensemble learning techniques are particularly effective for this task due to their ability to process complex datasets with high accuracy. Furthermore, incorporating explainable artificial intelligence into these predictions provides clarity and interpretability, allowing energy managers and homeowners to make informed decisions that optimize usage and reduce costs. This study comparatively analyzes decision tree–ensemble learning techniques augmented with explainable artificial intelligence for transparency and interpretability in residential building energy consumption forecasting. This approach employs the University Residential Complex and Appliances Energy Prediction datasets, data preprocessing, and decision-tree bagging and boosting methods. The superior model is evaluated using the Shapley additive explanations method within the explainable artificial intelligence framework, explaining the influence of input variables and decision-making processes. The analysis reveals the significant influence of the temperature-humidity index and wind chill temperature on short-term load forecasting, transcending traditional parameters, such as temperature, humidity, and wind speed. The complete study and source code have been made available on our GitHub repository at https://github.com/sodayeong for the purpose of enhancing precision and interpretability in energy system management, thereby promoting transparency and enabling replication.

accessible in the UCI Machine Learning Repository and can be found at https://archive.ics.uci.edu/dataset/374/appliances+energy+prediction (visited on 15 January 2024).

**Funding:** This research was supported by the MSIT (Ministry of Science and ICT), Korea, under the ICAN (ICT Challenge and Advanced Network of HRD) program (IITP-2024-2020-0-01832) supervised by the IITP (Institute of Information & Communications Technology Planning & Evaluation) and the Soonchunhyang University Research Fund. The funders had no role in study design, data collection and analysis, decision to publish, or preparation of the manuscript.

**Competing interests:** The authors have declared that no competing interests exist.

## Introduction

Integrating smart-grid technology in distribution, transmission, and production, particularly in smart homes, represents a transformative shift in energy management and consumption. As an emerging focus in smart-grid research, smart homes leverage information and communication technology or the Internet of Things to manage and automate residential environments [1–3], enabling residents to control devices remotely and monitor home status in real time, tailoring energy use to specific needs, such as scheduling electric heating [3]. Integral to these advancements is intelligent energy management, empowering homeowners to make informed decisions to reduce energy consumption [1, 3]. A pivotal aspect of this intelligent ecosystem is the home energy management system, which employs the Internet of Things and big data analytics to enhance energy efficiency in smart homes [4, 5]. In this realm, short-term load forecasting (STLF) emerges as a critical tool, facilitating energy trading and peak power management through effective energy strategies within a home energy management system [6]. However, achieving accurate STLF at the household level is complex, given the variability and uncertainty in the residential electricity demand, which is influenced by user behavior, weather conditions, and time variables.

Moreover, machine learning algorithms, such as artificial neural network-based deep learning and decision tree-based ensemble learning have been employed to address these challenges, modeling the intricate nonlinear relationships between energy consumption and its influencing factors [7]. Moon et al. [8] developed a short-term load forecasting model for a university campus that specifically segmented buildings into academic and dormitory complexes. Over a four-year period, daily electrical load data were collected and analyzed using a two-stage forecasting process incorporating moving averages and random forest (RF). The comparative analysis demonstrated that the proposed RF model outperformed other commonly used models, including decision tree, multiple linear regression (MLR), gradient boosting machine (GBM), support vector machine, and artificial neural network, in terms of predictive accuracy. Notably, Candanedo et al. [9] collected data on power consumption, smart sensor readings, and weather conditions. They developed and tested various forecasting models, including the GBM, RF, support vector machine, and MLR, with GBM exhibiting superior performance in household-level STLF. Their dataset, Appliances Energy Prediction (AEP), has become a valuable resource for the research community. However, while their study marked a significant milestone, it also highlighted areas for further improvement, such as the limitations of random sampling in time-series prediction, which can lead to challenges in generalization using out-of-sample data [10].

Building on previous advancements, recent research has increasingly focused on developing deep learning, especially hybrid deep learning models, for forecasting household electricity consumption using the AEP dataset. For instance, Munkhdalai et al. [11] introduced adaptive input selection (AIS) incorporated with a recurrent neural network (RNN), an end-to-end RNN model enhancing predictive accuracy. The AIS-RNN demonstrated superiority over various feature selection methods and machine learning algorithms. In addition, Yang et al [12] developed a fully convolutional model with an attention mechanism based on a deep residual network architecture rooted in convolutional neural networks (CNNs). This model is highlighted for its suitability in providing fast and accurate power consumption forecasting. Further, Sajjad et al. [13] and Khan et al. [14] expanded this domain by developing hybrid deep learning models combining CNNs with gated recurrent units (GRUs) and multilayer bidirectional GRUs, respectively, outperforming earlier models, such as the AIS-RNN and CNN-GRU. Furthering this trajectory, Bu and Cho [15] proposed a CNN and long short-term memory (LSTM) network model incorporating a multihead attention mechanism for

residential STLF, eclipsing traditional models (e.g., MLR, DT, and RF) on the Individual Household Electric Power Consumption dataset. Their work also introduced innovative visualizations through class activation maps. Similarly, Kiprijanovska et al. [16] developed a day-ahead STLF model based on a deep RNN, demonstrating superior performance over a range of traditional models, including MLR, DT, $k$-nearest neighbors, support vector machine, RF, GBM, and extreme gradient boosting (XGBoost), on the Pecan Street dataset covering around 1,000 residential buildings in Texas, USA. Their approach addressed the data shortage problem by deriving domain-specific time-series characteristics to train the model on household electricity consumption patterns effectively.

While deep learning models have demonstrated considerable success in residential building electricity consumption forecasting, it is essential to acknowledge instances where decision tree-based ensemble learning models have proven more effective. For instance, in their investigation into the STLF for multiple buildings, Bellahsen and Dagdougui [17] discovered that the RF model outperformed several deep learning models, including multilayer perceptron, LSTM, and sequence-to-sequence models. This superior performance, characterized by mean absolute percentage errors (MAPEs) between 1.67% and 4.80%, was attributed to the model's adept handling of diverse data. It is of particular importance to note that the use of explainable artificial intelligence (XAI) techniques, particularly Shapley additive explanations (SHAP), has the effect of highlighting the importance of certain variables in the RF model, thereby enhancing the transparency and reliability of the predictions. Further corroborating the potential superiority of ensemble learning models in certain scenarios, Zhang et al [18] utilized datasets from energy competitions hosted by Schneider Company, Kaggle, and ASHRAE to compare various forecasting models. The findings demonstrated that the light GBM (LightGBM) model outperformed traditional deep learning models, including bi-directional RNN, bi-directional LSTM (Bi-LSTM), and bi-directional GRU (Bi-GRU). The utilization of SHAP in this study also permitted a comprehensive analysis of the feature contributions, thereby corroborating the efficacy of the model. The findings of both studies indicate that, contingent upon the dataset and computing environment, decision tree-based ensemble learning models may occasionally exhibit superior performance relative to deep learning models in energy forecasting.

Building on this understanding, several studies have conducted in-depth performance analysis using SHAP to elucidate the interpretability of these decision tree-based ensemble learning models and to confirm their robustness in different energy management scenarios. Cui et al. [19] tailored their approach to U.S. residential buildings, using decision tree-based ensemble learning algorithms such as LightGBM and categorical boosting (CatBoost) to predict energy use intensity in different building types using the Residential Energy Consumption Survey dataset. Their SHAP analysis underscored the importance of factors such as total square footage and climatic conditions, highlighting how specific characteristics uniquely affect energy consumption in different building types. Meanwhile, Cakiroglu et al. [20] used ensemble learning to address several energy prediction challenges. One study focused on wind power prediction using a suite of models, including XGBoost and LightGBM, with data from Çanakkale, Turkey. SHAP analysis in this context identified wind speed as a critical predictive feature, improving the interpretability of the model and increasing its operational efficiency. In another study, Cakiroglu et al [21] applied these techniques to predict cooling loads in Malaysian buildings, where the CatBoost model excelled. SHAP identified the aspect ratio of buildings as a particularly influential feature, further informing design and energy efficiency strategies. These collective insights not only validate the efficacy of ensemble models in achieving high accuracy in energy predictions but also emphasize the value of SHAP in providing actionable insights, thus enhancing the transparency and applicability of predictive models in energy management.

This study acknowledges that there are instances where decision tree-based ensemble learning models may outperform deep learning models, particularly when variations in data quality and volume significantly impact model performance [22]. However, the existing literature reveals a significant gap in the rigorous comparative analysis of these models, especially when evaluated with XAI techniques such as local interpretable model-agnostic explanations (LIME) and SHAP [23, 24]. A comprehensive comparative analysis of decision tree-based ensemble learning and deep learning models is conducted, focusing on their application in forecasting energy consumption in both well-sampled environments such as university dormitories and poorly-sampled environments such as individual residential buildings. This approach not only evaluates the effectiveness of these models in real-world scenarios, but also employs a variety of XAI techniques to critically evaluate and highlight the strengths of SHAP in enhancing model transparency and interoperability, as well as to illustrate the limitations of traditional XAI methods [25]. By conducting a comprehensive comparison of the predictive performance and explainability of these AI models across a range of data conditions, a significant gap in the existing literature is addressed. The integration of multiple XAI methods provides a detailed assessment that enhances the transparency and reliability of energy forecasting models, thereby ensuring their practical applicability in energy management systems. Fig 1 depicts the architecture of the proposed method for load forecasting.

The main contributions of this paper are as follows:

- The proposed method is designed to effectively address data leakage by dynamically reconfiguring input variables to suit various environments, irrespective of the length of the data collection period or the known internal factors [26]. This ensures that the models are robust and applicable as benchmark models across diverse settings.

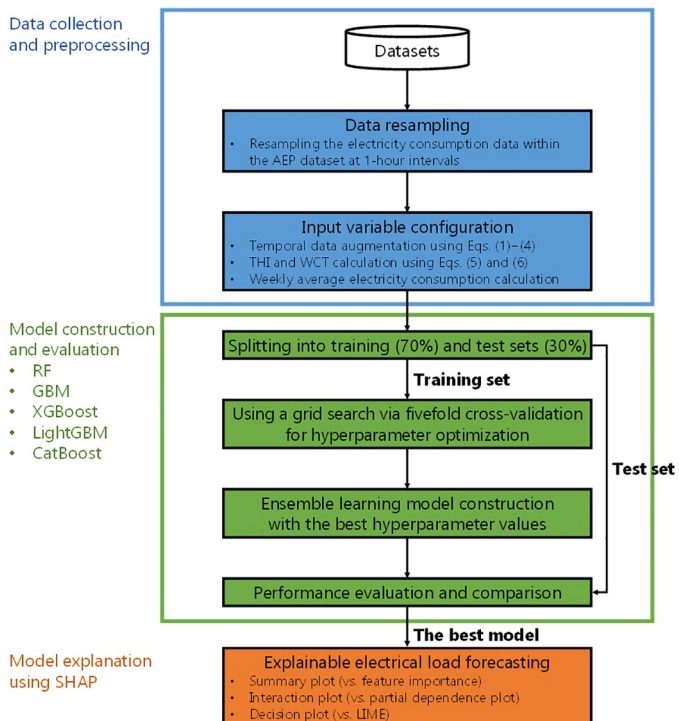

**Fig 1. Architecture of explainable residential building electrical energy consumption modeling.**

- To assess the efficacy of the proposed method, we conducted an extensive comparative study involving five decision tree-based ensemble learning models and ten deep learning models. This analysis compares the efficacy of ensemble learning models to that of deep learning counterparts, providing a deeper insight into their potential and performance under varied conditions.

- The performance of these models is dissected not only by consumption timing—covering weekdays, holidays, and weekends—but also by examining the influence of internal and external factors combined. This dual perspective approach enriches the understanding of model adaptability and accuracy in real-world scenarios.

- In addition to SHAP, a range of XAI techniques are employed and compared in order to elucidate the decision-making processes of our models. This comparative study highlights the effectiveness of decision tree-based ensemble learning models in interpreting complex datasets such as the AEP, and extends to sufficiently sampled university dormitory datasets, broadening the scope of analysis.

- The use of partial dependence plots (PDPs) to examine the holiday indicator offers insights into how telecommuting patterns could influence energy management strategies. This analysis assists home energy management system managers in developing more effective operational strategies by capitalizing on behavioral tendencies during non-standard days.

The paper is structured as follows. The Forecasting model development section explores data preprocessing for training decision tree-based ensemble learning models and outlines the construction of the proposed model. Then, the Experimental results and discussion section presents the results of the experiments and discusses their implications. Finally, The Conclusions section offers conclusions and directions for future research.

## Forecasting model development

### Data gathering and preparation for model development

We aimed to develop an explainable electricity consumption prediction model for residential buildings, encompassing both complex and individual structures. Our study first utilized the University Residential Complex dataset, which comprises 40 months of electricity consumption data from 16 dormitory buildings, collected from September 1, 2015, to December 31, 2018, at a private university in Seoul, South Korea. This dataset includes hourly measurements obtained from the Korea Electric Power Corporation (KEPCO) Power Planner system, continuously monitored in real-time by the university's energy security team. Additionally, we employed regional synoptic meteorological data from the Korea Meteorological Administration (KMA) for the same period to assess the influence of weather conditions on electricity consumption.

To improve the accuracy of our analysis, we integrated the publicly available AEP dataset [8], which originally recorded electricity consumption at 10-minute intervals. To ensure alignment with the University Residential Complex dataset, we resampled the AEP data to hourly intervals. This resampling not only facilitates direct comparison between the two datasets but also leverages the accessibility and comprehensiveness of the AEP dataset, thereby increasing the robustness and applicability of our energy consumption predictions. This approach also aligns the dataset with realistic household energy consumption patterns, making it highly relevant to practical energy management, and simplifies data analysis, enabling the proposed model to efficiently forecast day-ahead electricity consumption on an hourly basis [27–29].

**Table 1. Statistical values and key attributes for residential building energy datasets.**

| Statistics | University Residential Complex (unit: kWh) | Appliances Energy Prediction (unit: Wh) |
|---|---|---|
| Minimum | 570.2 | 170.0 |
| First quartile | 993.2 | 300.0 |
| Median | 1174.0 | 380.0 |
| Mean | 1217.6 | 586.2 |
| Third quartile | 1392.5 | 660.0 |
| Maximum | 2588.0 | 4000.0 |
| Collection duration | September 1, 2015–December 31, 2018 | January 11, 2016–May 27, 2016 |
| Building location | Seoul, Republic of Korea | Stambruges, Belgium |
| Building types and counts | 16 dormitory buildings | One private home |
| Public access | No | Yes |

Both datasets are comprehensively summarized in Table 1, which details their statistical values and key attributes.

In order to address the challenge of data leakage, which is a common problem in real-time predictive modeling, we carefully reconstructed the input variables [9]. This issue is particularly relevant when using sensor data from the AEP dataset, as these values may not be available at the prediction point in real-time applications, making them difficult to use directly in the energy industry. In order to reconstruct the variables, we employed historical electricity consumption data as an internal factor, while timestamps and weather information were incorporated as external factors [30]. The integration of these elements ensures the model's reliability in predicting household electrical energy consumption by considering both internal behavioral patterns and external environmental factors. This approach results in a robust tool for effective energy management in residential settings.

Timestamp data, including markers for holidays, days of the week (DOTW), hours, and months, are instrumental in uncovering distinct power usage patterns within buildings [22]. For example, hourly energy consumption may fluctuate in response to mealtimes, business hours, and other variables. Furthermore, the recognition of holidays and weekdays can explain contrasting consumption trends on weekends versus weekdays. However, the limitation of a one-dimensional format in capturing the cyclical nature of time necessitates the transition to a two-dimensional representation. This transformation allows for a more accurate interpretation of the sequential and cyclical patterns in time variables, such as the transition from day to night or across different days of the week. Consequently, acknowledging the sequential nature of hours and days, we implemented Eqs (1) to (4) to transform the format of these categorical elements from a one- to two-dimensional continuum, effectively capturing their cyclical nature [22]:

$$Hour_x = \sin(360°/24 \times Hour) \tag{1}$$

$$Hour_y = \cos(360°/24 \times Hour) \tag{2}$$

$$DOTW_x = \sin(360°/7 \times DOTW) \tag{3}$$

$$DOTW_y = \cos(360°/7 \times DOTW) \tag{4}$$

where *Hour* denotes the hour of the day.

In the AEP dataset, we assigned a value of 0 to weekdays (Monday through Friday) and 1 to weekends (Saturday and Sunday) using the "*WeekStatus*" variable from the original AEP dataset [9]. Although the month also has a crucial role in determining electricity consumption patterns, the limitations in the AEP dataset duration (not spanning beyond a year) precluded incorporating the monthly trends in the analysis. Similarly, to ensure consistency in our analysis across different datasets, we also excluded month-related variables from the University Residential Complex dataset. Additionally, we extracted holiday information from Time and Date (https://www.timeanddate.com/) to accurately account for variations in energy usage on public holidays in both datasets. This approach ensures a uniform treatment of temporal variables, enhancing the comparability and robustness of our proposed model.

The significance of weather conditions in predicting electricity consumption cannot be overstated, as the use of high-energy appliances, such as heaters and air conditioners, is intricately linked with climatic variations [31]. The AEP dataset encompasses six meteorological parameters: temperature, humidity, wind speed, atmospheric pressure, visibility, and dew point. To augment the applicability of the proposed methodology in real-life settings, we prioritized temperature, humidity, and wind speed as the primary input factors. These elements exhibit a pronounced correlation with electricity usage patterns [30, 32]. For the University Residential Complex dataset, we similarly focused only on these parameters, using reliable short-term forecasts from the KMA. This strategy was employed to avoid data leakage and to enhance the practical applicability of our model in the energy industry.

Despite the presented data, it was challenging to discern a distinct relationship between humidity/wind speed and electricity consumption. To establish a more direct link to power consumption, we calculated the WCT and the THI, also known as the discomfort index. These indices, which incorporate temperature, humidity, and wind speed, reflect the perceived environmental comfort or stress during summer and winter, thereby providing a more relevant measure of weather impact on power consumption [33, 34]. As described in Eqs (5) and (6), *Temp* represents temperature, *Humi* signifies humidity, and *WS* denotes wind speed. Consequently, by integrating these comprehensive indicators, we incorporated 10 input variables as external factors in developing the STLF model:

$$THI = (1.8 \times Temp + 32) - [(0.55 - 0.0055 \times Humi) \times (1.8 \times Temp - 26)] \quad (5)$$

$$WCT = 13.12 + 0.6215 \times Temp - 11.37 \times WS^{0.16} + 0.3965 \times Temp \times WS^{0.16} \quad (6)$$

We employed historical electricity consumption data as internal factors because they reflect the recent power consumption trends. Thus, to ensure our model captures temporal dynamics, we used the power consumption data for one day and one week prior to the prediction time point [17]. This approach allows the model to incorporate the most recent usage patterns, providing it with a temporal context that is crucial for accurate forecasting. The power consumption data for the week prior can reveal the most recent trends for the DOTW, whereas the power consumption data for the day prior can indicate the most recent trends for the hour of the day. In addition, since power consumption patterns for holidays and weekdays differ significantly, we added indicators for holidays to accurately distinguish between these two types of power consumption data [30, 32].

To capture the power consumption trend effectively for the prediction time point during a week, we incorporated historical power consumption data as an internal factor. We differentiated the input based on whether the prediction time point was a holiday or a weekday. Specifically, if the prediction time point was a holiday, we calculated the average power consumption for holidays over the previous seven days. Similarly, if the prediction time point was a weekday,

**Table 2. Selected input variables, descriptions, and types.**

| Variables | Description | Data Type |
|---|---|---|
| $Hour_x$ | Hour of the day (sine value) | Timestamp (numeric) |
| $Hour_y$ | Hour of the day (cosine value) | Timestamp (numeric) |
| $DOTW_x$ | Day of the week (sine value) | Timestamp (numeric) |
| $DOTW_y$ | Day of the week (cosine value) | Timestamp (numeric) |
| $Holi$ | Weekday: 0; weekend: 1 | Timestamp (binary) |
| $Temp$ | Hourly temperature | Weather condition (numeric) |
| $Humi$ | Hourly humidity | Weather condition (numeric) |
| $WS$ | Hourly wind speed | Weather condition (numeric) |
| $THI$ | Hourly temperature-humidity index | Weather condition (numeric) |
| $WCT$ | Hourly wind chill temperature | Weather condition (numeric) |
| $Cons_1$ | Power consumption one day before the time point | Historical electricity consumption (numeric) |
| $Holi_1$ | Holiday indicator one day before the time point | Historical electricity consumption (binary) |
| $Cons_7$ | Power consumption one week before the time point | Historical electricity consumption (numeric) |
| $Holi_7$ | Holiday indicator one week before the time point | Historical electricity consumption (binary) |
| $Cons_{avg}$ | Weekly average power consumption | Historical electricity consumption (numeric) |

we calculated the average power consumption for weekdays over the same period. This strategy allows us to train the model to reflect typical consumption patterns for each type of day, thereby increasing predictive accuracy. Thus, we identified five input variables as internal factors to construct the STLF model. Table 2 summarizes all input variables and their information.

## Comprehensive guide to ensemble learning methods

Ensemble learning methods capitalize on the collective strength of multiple learning algorithms to achieve superior predictive performance that would be unattainable through the use of any individual learning algorithm [10, 22]. The fundamental principle underlying ensemble methods is the integration of diverse models to minimize variance, bias, or enhance predictions [32]. There are several primary strategies for constructing ensemble models, including:

- Bagging: The objective of bagging, or bootstrap aggregating, is to reduce the variance of a prediction model by generating additional data through bootstrapping, which is random sampling with replacement [35, 36]. These additional data are then used to train multiple models on different datasets. The RF algorithm is a classic example of a bagging method, where multiple decision trees are trained independently and their predictions are averaged to produce the final result [8, 36, 37]. This approach enhances the model's robustness and accuracy by mitigating overfitting. We selected RF due to its proven efficacy in managing large datasets and its robustness against overfitting.

- Boosting: The objective of boosting is to reduce the bias of the model by sequentially training weak models, each compensating for the weaknesses of its predecessors [38]. The most prominent boosting algorithms include GBM, XGBoost, LightGBM, and CatBoost [32, 38]. These methods iteratively improve the model by minimizing a specified loss function, with each new model trained to correct the errors of the previous models. This process results in the creation of a robust ensemble that is capable of modelling intricate relationships and achieving high levels of predictive accuracy. Boosting methods were chosen for their ability to incrementally correct errors and handle nonlinear relationships effectively.

- Stacking: The process of stacking involves training multiple models (base learners) and then combining their outputs using a meta-learner [39, 40]. The meta-learner is trained to optimally integrate the predictions from the base learners, typically using a holdout dataset. This method allows for the capture of a wider range of patterns and interactions by leveraging the strengths of various models. Stacking was included to explore how combining different model outputs could enhance predictive accuracy.

The domain of decision tree-based ensemble learning techniques is among the most reliable supervised learning methodologies for regression and classification tasks [41, 42]. These techniques are instrumental in developing forecasting models and are notable for their ease of interpretation, robust performance, and high precision [43]. Their efficacy in predicting energy consumption has been demonstrated across energy forecasting sectors [44, 45].

Furthermore, decision tree-based ensemble learning methods surpass traditional statistical approaches, such as MLR, in their capacity to encompass a wide array of machine learning concepts and adeptly model nonlinear relationships. In pursuit of accurate STLF at the residential building level with a 1-h interval, we employed five distinguished decision tree-based ensemble learning approaches: RF, GBM, XGBoost, LightGBM, and CatBoost. Table 3 delineates the advantages and disadvantages of these decision tree-based ensemble learning methodologies [46–51].

- RF [46, 47] is an ensemble learning method that constructs multiple decision trees during training and outputs the average prediction of the individual trees. Each tree in the forest is constructed from a bootstrap sample of the training data and uses a random subset of features for splitting nodes. This randomization helps to reduce overfitting and increase the robustness of the model. RF is highly effective for large datasets with high dimensionality and has excellent performance in terms of accuracy and interpretability.

**Table 3. Advantages and disadvantages of decision tree–based ensemble learning methods.**

| Methods | Advantages | Disadvantages |
|---|---|---|
| Random forest (RF) [46, 47] | • Excels at efficiently managing a multitude of independent variables without variable elimination<br>• Delivers robust predictive performance with minimal hyperparameter tuning | • Performance may diminish with extremely high-dimensional sparse data, although it is competent for most high-dimensional datasets |
| Gradient boosting machine (GBM) [48, 49] | • Provides excellent predictive outcomes for complex regression tasks, including ranking and Poisson regression<br>• Offers extensive hyperparameter tuning capabilities, enhancing model flexibility and precision | • Time-intensive model development due to the sequential nature of boosting<br>• Careful tuning is required due to the overfitting risk, particularly with outliers |
| Extreme gradient boosting (XGBoost) [49, 50] | • Implements regularization to mitigate overfitting and has a faster training process than GBM and RF<br>• Achieves high predictive accuracy across diverse datasets, including those with complex structures and various subgroupings | • While designed for sparse data, performance may vary; careful hyperparameter tuning is essential to prevent overfitting |
| Light gradient boosting machine (LightGBM) [49, 50] | • Outperforms many other boosting algorithms in terms of speed, efficiency, and accuracy<br>• Enables parallel and graphics processing unit learning, optimizing resource utilization<br>• Exhibits lower memory requirements, enhancing scalability | • May be prone to overfitting on small datasets if not appropriately regularized |
| Categorical boosting (CatBoost) [51] | • Features automatic preprocessing for categorical variables, streamlining data preparation<br>• Provides robust predictive accuracy with minimal hyperparameter tuning, facilitating ease of use | • Performs well on various data types, but training speed might vary with the complexity and structure of the dataset |

- GBM [48, 49] builds an ensemble of trees in a sequential manner, with each tree attempting to correct the errors of the previous one. This method focuses on optimizing a loss function by adding new models that minimize this loss. GBM is highly flexible and can be adapted to various loss functions, making it suitable for both regression and classification tasks. The iterative nature of GBM allows it to learn complex patterns and interactions within the data.

- XGBoost [49, 50] is an advanced implementation of gradient boosting that incorporates several optimizations to enhance performance and efficiency. These include regularization to prevent overfitting, parallel processing to speed up computation, and handling of missing values. XGBoost also employs a more efficient algorithm for finding optimal splits in the trees. These enhancements render XGBoost particularly effective for large-scale datasets and complex predictive modeling tasks.

- LightGBM [49, 50] represents a significant advancement over traditional boosting methods. It employs a leaf-wise tree growth strategy and a histogram-based decision tree algorithm. These innovations have the effect of significantly reducing memory usage and training time, thereby rendering LightGBM highly efficient and scalable. It is well-suited for large datasets with many features and provides strong predictive performance with faster computation.

- The CatBoost algorithm [51] is designed to handle categorical features more effectively than other algorithms. It employs a special technique called "ordered boosting" to reduce target leakage and prediction shift. CatBoost automatically preprocesses categorical variables and uses a combination of categorical and numerical features in a balanced manner. This results in high accuracy and robust performance, especially in datasets with a significant number of categorical variables.

Hyperparameters were optimized through a five-fold cross-validation process within the training dataset to refine the performance of the decision tree-based ensemble learning methodologies. The search for optimal hyperparameter settings was conducted with the aid of GridSearchCV, a utility that performs an exhaustive search over specified hyperparameter values within the scikit-learn library [41, 52, 53] as described in Table 4.

## Enhancing interpretability of ensemble learning models

This study expounds on the application of decision tree-based ensemble learning models employing SHAP [25, 54], elucidating the significance and influence of input variables and enhancing the interpretability of the model. Primarily leveraged for model transparency within decision tree-based ensemble learning frameworks, this methodology also proposes potential insight. Furthermore, the Tree SHAP technique enables a coherent interpretation of the decision tree-based ensemble learning model training outcomes, which is critical for understanding household electricity consumption forecasts.

Employing the SHAP technique conveys the valuation of input variables by computing Shapley values for each predictor, delineating their influence on the response variable. This process is pivotal in clarifying the nature of AI models and the foundations of their learning outcomes [19–21]. Additionally, the SHAP technique facilitates the explanation of individual predictions by ascertaining the Shapley value for each predictor, permitting a precise understanding of their effects on machine learning model predictions. A global interpretation is attainable by aggregating these results.

A quantitative assessment of each predictor's contribution is accomplished via Shapley values, encapsulating the negative and positive influences. For instance, when setting the power

**Table 4. List of hyperparameters for decision tree-based ensemble learning method.**

| Methodologies | References | Hyperparameters |
|---|---|---|
| Random forest | [52] | • Trees count: 128<br>• Features per split: auto, sqrt, log2 |
| Gradient boosting machine | [41] | • Iteration count: 100, 250, 500<br>• Learning rate: 0.01, 0.05, 0.1<br>• Depth: 5, 10<br>• Loss type: quantile, Huber |
| Extreme gradient boosting | [41] | • Iterations: 250, 500, 1000<br>• Learning rate: 0.01, 0.05, 0.1<br>• Depth: 6, 8, 10<br>• Subsampling rate: 0.5, 0.75, 1.0<br>• Feature sample by tree/level/node: 0.5, 0.75, 1.0<br>• Booster type: gbdt, dart |
| Light gradient boosting machine | [41] | • Iterations: 1000, 1500<br>• Learning rate: 0.01, 0.05, 0.1<br>• Leaves: 64<br>• Subsample: 0.5<br>• Feature sample by tree: 1.0<br>• Booster type: gbdt, dart |
| Categorical boosting | [53] | • Learning rate: 0.03, 0.1<br>• Max tree depth: 4, 6, 10<br>• L2 regularization levels: 1, 3, 5, 7, 9 |

usage as the dependent variable, a substantial positive Shapley value for temperature implies that an increase in temperature correlates with an increase in power usage. The Shapley value, delineating the weight and contribution of each predictor to the outcome, is computed as follows:

$$\Phi_i = \sum_{S \subseteq F \setminus \{i\}} (|S|! \times (|F| - |S| - 1)!) / (|F|!) \times [f_{S \cup i}(x_{S \cup i}) - f_S(x_S)]. \tag{7}$$

Eq 7 calculates $\Phi_i$, symbolizing the contribution of the independent variable $i$ to the ensemble of predictors ($F$), with $f$ denoting the predictive model. This contribution is ascertained by evaluating the discrepancy when variable $i$ is included versus when it is excluded from the subset ($S$) of predictors, where the condition $S \subseteq F$ is satisfied. The weighting accorded to each subset $S$ is obtained by averaging the factorial of the total permutations of set $F$ ($|F|!$), subset $S$ ($|S|!$), and the set excluding $i$ and $S$ ($|F| - |S| - 1)!$).

## Performance metrics

In the empirical analysis, the efficacy of the decision tree-based ensemble learning models was assessed using three statistical metrics: MAPE, CVRMSE, and NMAE. These metrics are advantageous because they convey accuracy in terms of percentage error, rendering the results transparent for technical and nontechnical stakeholders. The following equations were applied

to calculate MAPE, CVRMSE, and NMAE:

$$MAPE = 1/n \times \sum \left| (Y_t - \hat{Y}_t)/Y_t \right| \times 100 \tag{8}$$

$$CVRMSE = (\sqrt{(1/n \times \sum (Y_t - \hat{Y}_t)^2)})/\ddot{Y} \times 100 \tag{9}$$

$$NMAE = \left( 1/n \times \sum |Y_t - \hat{Y}_t| \right)/\ddot{Y} \times 100. \tag{10}$$

In these equations, at any given time $t$, $Y_t$ denotes the actual observed value, whereas $\hat{Y}_t$ signifies the model prediction. The symbol $\ddot{Y}$ represents the mean of all observed values. The variable $n$ corresponds to the total number of predictive instances, and $t$ indexes a specific temporal prediction point.

In addition, we calculated the harmonic mean (HM) by considering these three metrics to represent the accuracy of decision tree-based ensemble learning models quantitatively. The harmonic mean is particularly well-suited to the combination of disparate error metrics, as it effectively mitigates the impact of extreme outliers while providing a balanced measure of performance. This approach is particularly advantageous when dealing with metrics that can vary significantly in magnitude. The *HM* is calculated in Eq (11):

$$HM = 3/((1/MAPE) + (1/CVRMSE) + (1/NMAE)). \tag{11}$$

The harmonic mean is employed to provide a more comprehensive and robust evaluation of model performance. This method ensures that models which perform well across all three metrics are favored, rather than models that might excel in one metric but perform poorly in others. This comprehensive approach to model assessment supports a balanced interpretation of the results, making it easier to identify the most reliable and accurate models.

## Experimental results and discussion

In order to conduct this study, the datasets were adjusted to reflect a stratification into 70% training and 30% testing subsets. For the University Residential Complex dataset, the training period was defined as spanning from September 1, 2015, to December 31, 2017, which comprises 28 months or approximately 70% of the data. The testing period encompassed the period from January 1, 2018, to December 31, 2018, which is 12 months, representing the remaining 30%. To synchronize the timelines across datasets, the AEP hourly electricity consumption dataset was also divided accordingly. The training period extended from January 11, 2016, at 6:00 p.m. to April 16, 2016, encompassing approximately 70% of the data. The testing period was scheduled to commence on April 17, 2016, and conclude on May 27, 2016, at 6:00 p.m., incorporating the final 30% of the dataset.

### Comparison of decision tree-based ensemble learning models

The Taylor diagram in Fig 2 provides a comprehensive visual assessment of different ensemble learning techniques, illustrating their performance in terms of correlation with observed data and centered RMSE. Most ensemble models show correlations above 0.9, indicating high accuracy in capturing data patterns. A few models approach a correlation of 0.95, highlighting their superior predictive accuracy and low RMSE values, demonstrating their reliability. The clustering of the models near the observed benchmark indicates their effectiveness, as they are close to the ideal performance point with high correlation and low RMSE. Overall, the

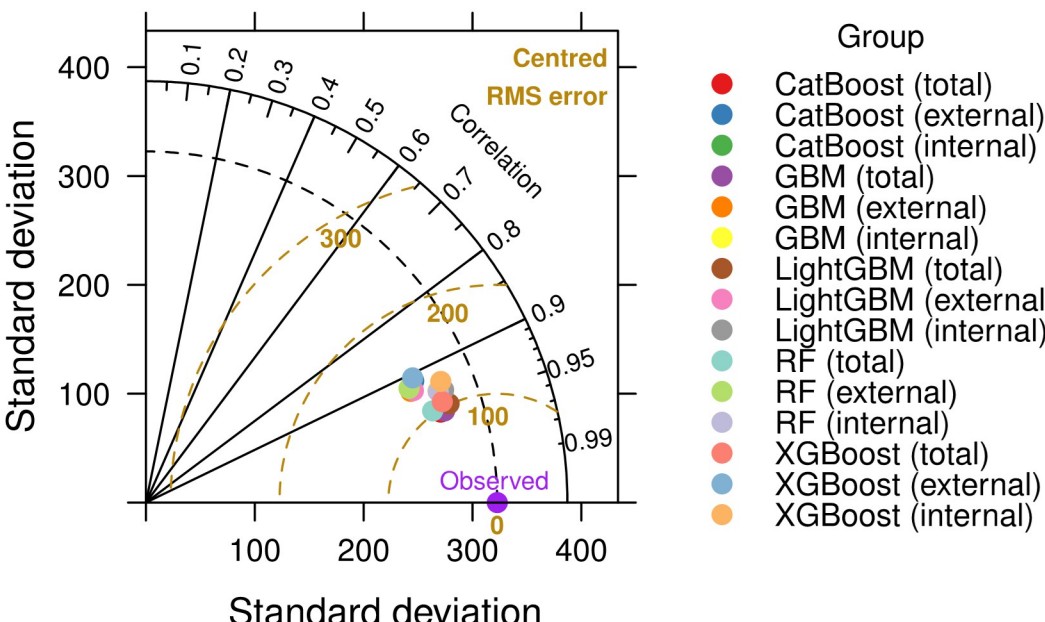

**Fig 2. Taylor diagram showing the performance of ensemble learning models on the University Residential Complex dataset.**

evaluated ensemble learning models are highly effective, with most showing exceptional accuracy and reliability, making them invaluable for applications requiring precision.

The Taylor diagram in Fig 3 illustrates the performance of various ensemble learning models, focusing on the RF model trained on a large dataset. Although a moderate correlation of about 0.6 was achieved, a comparison with Fig 2 indicates areas for improvement, especially

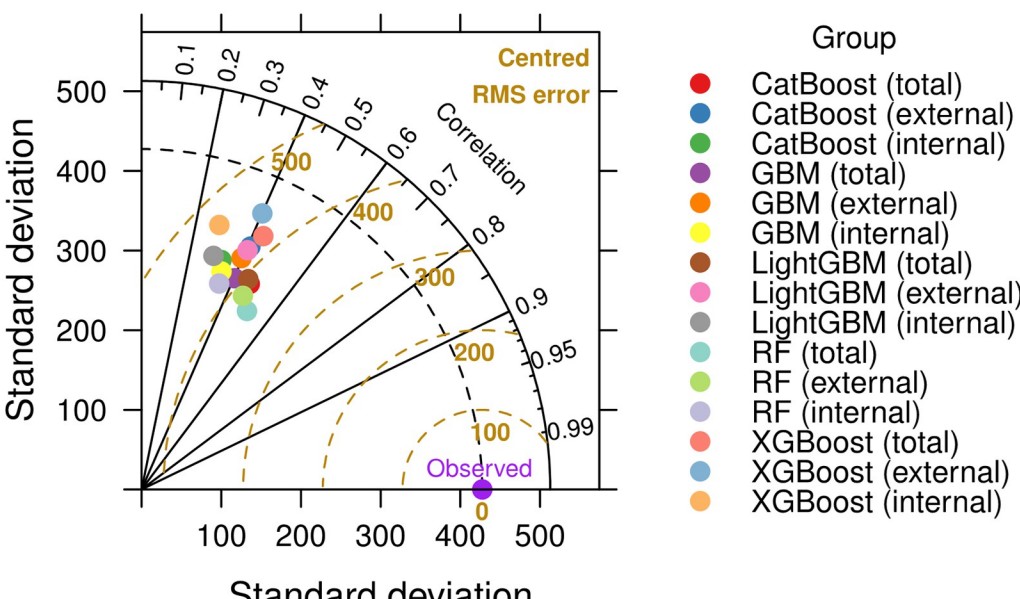

**Fig 3. Taylor diagram showing the performance of ensemble learning models on the Appliances Energy Prediction dataset.**

when moving from large to small-scale applications. The performance of the RF model in predicting energy consumption for individual devices highlights challenges such as increased variability and noise in small-scale data, the need for large amounts of data to make accurate predictions, and the need for customized features to capture specific device characteristics. These factors underscore the difficulty of applying models trained on large datasets to smaller, more variable datasets.

Tables 5–10 highlight the comparative analyses of MAPE, CVRMSE, and NMAE across input variable configurations delineated by date type within the AEP dataset. The subscripts

**Table 5. Comparative analysis of decision tree–based ensemble learning models trained with external factors on the University Residential Complex dataset.** All evaluation metrics are presented in percentage (%).

| Models | $MAPE_W$ | $MAPE_H$ | $MAPE_T$ | $CVRMSE_W$ | $CVRMSE_H$ | $CVRMSE_T$ | $NMAE_W$ | $NMAE_H$ | $NMAE_T$ |
|---|---|---|---|---|---|---|---|---|---|
| RF | 11.14 | 12.37 | 11.54 | 14.33 | 16.83 | 15.05 | 11.64 | 13.16 | 12.07 |
| GBM | 11.43 | 12.42 | 11.75 | 14.48 | 16.70 | 15.12 | 11.90 | 13.19 | 12.27 |
| XGBoost | 11.44 | 12.78 | 11.88 | 14.61 | 17.32 | 15.39 | 11.86 | 13.50 | 12.33 |
| LightGBM | 11.25 | 12.45 | 11.64 | 14.24 | 16.72 | 14.95 | 11.70 | 13.20 | 12.13 |
| CatBoost | 11.49 | 12.66 | 11.87 | 14.63 | 17.02 | 15.32 | 11.92 | 13.38 | 12.34 |

**Table 6. Comparative analysis of decision tree–based ensemble learning models trained with internal factors on the University Residential Complex dataset.** All evaluation metrics are presented in percentage (%).

| Models | $MAPE_W$ | $MAPE_H$ | $MAPE_T$ | $CVRMSE_W$ | $CVRMSE_H$ | $CVRMSE_T$ | $NMAE_W$ | $NMAE_H$ | $NMAE_T$ |
|---|---|---|---|---|---|---|---|---|---|
| RF | 5.24 | 7.77 | 6.06 | 7.64 | 11.22 | 8.71 | 5.41 | 7.98 | 6.14 |
| GBM | 5.16 | 7.66 | 5.98 | 7.68 | 11.22 | 8.73 | 5.35 | 7.84 | 6.06 |
| XGBoost | 5.58 | 8.39 | 6.50 | 7.95 | 12.19 | 9.23 | 5.70 | 8.62 | 6.54 |
| LightGBM | 5.11 | 7.71 | 5.96 | 7.39 | 11.40 | 8.60 | 5.24 | 7.94 | 6.01 |
| CatBoost | 5.14 | 7.74 | 5.99 | 7.42 | 11.32 | 8.60 | 5.27 | 8.03 | 6.06 |

**Table 7. Comparative analysis of decision tree–based ensemble learning models trained with external and internal factors on the University Residential Complex dataset.** All evaluation metrics are presented in percentage (%).

| Models | $MAPE_W$ | $MAPE_H$ | $MAPE_T$ | $CVRMSE_W$ | $CVRMSE_H$ | $CVRMSE_T$ | $NMAE_W$ | $NMAE_H$ | $NMAE_T$ |
|---|---|---|---|---|---|---|---|---|---|
| RF | 4.94 | 6.35 | 5.40 | 7.47 | 9.99 | 8.20 | 5.24 | 6.81 | 5.69 |
| GBM | 4.69 | 6.02 | 5.13 | 7.01 | 9.30 | 7.67 | 4.93 | 6.38 | 5.35 |
| XGBoost | 4.84 | 6.75 | 5.46 | 7.21 | 10.57 | 8.21 | 5.09 | 7.14 | 5.68 |
| LightGBM | 4.65 | 6.46 | 5.24 | 6.88 | 10.10 | 7.84 | 4.86 | 6.83 | 5.43 |
| CatBoost | 4.73 | 5.95 | 5.13 | 7.14 | 9.56 | 7.85 | 5.01 | 6.39 | 5.40 |

**Table 8. Comparative analysis of decision tree–based ensemble learning models trained with external factors on the Appliances Energy Prediction dataset.** All evaluation metrics are presented in percentage (%).

| Models | $MAPE_W$ | $MAPE_H$ | $MAPE_T$ | $CVRMSE_W$ | $CVRMSE_H$ | $CVRMSE_T$ | $NMAE_W$ | $NMAE_H$ | $NMAE_T$ |
|---|---|---|---|---|---|---|---|---|---|
| RF | 49.73 | 34.45 | 45.60 | 68.55 | 68.14 | 68.61 | 46.76 | 39.91 | 44.68 |
| GBM | 45.70 | 35.71 | 43.01 | 73.26 | 74.05 | 73.75 | 45.90 | 42.92 | 44.99 |
| XGBoost | 49.99 | 39.31 | 47.11 | 78.40 | 77.81 | 78.43 | 48.77 | 44.09 | 47.35 |
| LightGBM | 49.61 | 36.73 | 46.13 | 73.22 | 75.02 | 74.06 | 48.17 | 43.51 | 46.75 |
| CatBoost | 50.70 | 32.85 | 45.88 | 75.44 | 71.61 | 74.38 | 49.56 | 40.46 | 46.79 |

**Table 9. Comparative analysis of decision tree–based ensemble learning models trained with internal factors on the Appliances Energy Prediction dataset.** All evaluation metrics are presented in percentage (%).

| Models | $MAPE_W$ | $MAPE_H$ | $MAPE_T$ | $CVRMSE_W$ | $CVRMSE_H$ | $CVRMSE_T$ | $NMAE_W$ | $NMAE_H$ | $NMAE_T$ |
|---|---|---|---|---|---|---|---|---|---|
| RF | 49.64 | 38.74 | 46.70 | 76.26 | 69.01 | 74.09 | 47.59 | 41.69 | 45.80 |
| GBM | 43.26 | 36.75 | 41.50 | 75.99 | 72.60 | 75.07 | 44.22 | 41.93 | 43.52 |
| XGBoost | 54.58 | 46.35 | 52.36 | 85.79 | 75.74 | 82.75 | 52.95 | 47.77 | 51.37 |
| LightGBM | 54.36 | 44.15 | 51.60 | 80.90 | 74.48 | 79.00 | 51.45 | 46.25 | 49.87 |
| CatBoost | 51.87 | 40.45 | 48.79 | 80.25 | 70.06 | 77.16 | 49.59 | 43.35 | 47.69 |

**Table 10. Comparative analysis of decision tree–based ensemble learning models trained with external and internal factors on the Appliances Energy Prediction dataset.** All evaluation metrics are presented in percentage (%).

| Models | $MAPE_W$ | $MAPE_H$ | $MAPE_T$ | $CVRMSE_W$ | $CVRMSE_H$ | $CVRMSE_T$ | $NMAE_W$ | $NMAE_H$ | $NMAE_T$ |
|---|---|---|---|---|---|---|---|---|---|
| RF | 49.62 | 36.72 | 46.14 | 66.62 | 65.04 | 66.28 | 45.82 | 39.62 | 43.94 |
| GBM | 43.20 | 32.71 | 40.37 | 72.65 | 69.88 | 71.93 | 43.96 | 39.62 | 42.64 |
| XGBoost | 57.33 | 40.12 | 52.68 | 78.00 | 70.95 | 75.89 | 52.16 | 42.81 | 49.32 |
| LightGBM | 48.56 | 33.11 | 44.39 | 69.25 | 69.77 | 69.63 | 45.78 | 40.05 | 44.04 |
| CatBoost | 44.80 | 32.04 | 41.35 | 69.15 | 67.20 | 68.69 | 44.51 | 38.28 | 42.61 |

W, H, and T in these tables indicate weekdays, holidays, and all (total) days, respectively. A discernible pattern emerged from this comparison: the statistical values attained their nadir when external factors were considered.

The results from the University Residential Complex dataset indicate that models demonstrated consistent performance with external factors, with MAPE values between 11–12% for both weekdays and holidays, with slight variations among models. However, CVRMSE values indicated increased error variability during holidays. Significantly superior outcomes were observed when models were trained exclusively with internal factors. For example, RF model's $MAPE_W$ decreased from 11.14% with external factors to 5.24%, demonstrating the pivotal role of internal data in reducing predictive errors. Similar enhancements were observed in CVRMSE and NMAE, indicating enhanced precision and accuracy. The integration of both external and internal factors yielded the most optimal performance across all metrics, suggesting that this combined approach effectively leverages the strengths of both datasets. As demonstrated by Table 11, harmonic mean comparison corroborates this assertion. The integrated approach exhibited the lowest error values, thereby substantiating the efficacy of integrating both external and internal factors.

The analysis of the Appliances Energy Prediction dataset revealed that training with external factors resulted in higher error ranges, with MAPE figures exceeding 45% for all models,

**Table 11. Harmonic mean comparison considering various input variable configurations on the University Residential Complex dataset, presented as percentages (%).**

| Models | External | Internal | Total |
|---|---|---|---|
| Random forest | 12.72 | 6.78 | 6.21 |
| Gradient boosting machine | 12.89 | 6.71 | 5.86 |
| Extreme gradient boosting | 13.03 | 7.22 | 6.24 |
| Light gradient boosting machine | 12.75 | 6.66 | 5.97 |
| Categorical boosting | 13.01 | 6.69 | 5.91 |
| Average | 12.88 | 6.81 | 6.04 |

**Table 12. Harmonic mean comparison considering various input variable configurations on the Appliances Energy Prediction dataset, presented as percentages (%).**

| Models | External | Internal | Total |
|---|---|---|---|
| Random forest | 50.95 | 52.87 | 50.40 |
| Gradient boosting machine | 50.82 | 49.68 | 48.29 |
| Extreme gradient boosting | 54.45 | 59.23 | 57.21 |
| Light gradient boosting machine | 53.03 | 57.59 | 50.34 |
| Categorical boosting | 52.99 | 55.12 | 48.23 |
| Average | 52.45 | 54.90 | 50.89 |

indicating greater variability and potential issues with overfitting or underfitting. Conversely, when models were trained with internal factors, the high error rates were somewhat mitigated, particularly in GBM and CatBoost models, which exhibited superior performance in handling internal dataset characteristics. However, the most significant improvement in performance metrics was observed when both external and internal factors were combined. This combined approach effectively reduced errors across all models, as demonstrated by Table 10 and the harmonic mean comparison in Table 12. For instance, as illustrated in Table 10, CatBoost's $MAPE_T$ improved to 41.35%, and RF's $CVRMSE_T$ reduced to 66.28%. Moreover, Table 12 illustrates that CatBoost's harmonic mean improved to 48.23%, while GBM's harmonic mean reduced to 48.29%. These results provide further evidence of the efficacy of the combined approach.

The utilization of decision tree-based ensemble models in a combined training approach has demonstrated substantial adaptability and accuracy, rendering them suitable for complex real-world applications such as energy prediction and residential management systems. The GBM model demonstrated superior performance on the University Residential Complex dataset, while the CatBoost model exhibited superior performance on the AEP dataset. These findings suggest that a balanced integration of internal and external data sources may be beneficial for optimizing model performance across various conditions.

## Comparison with state-of-the-art deep learning and hybrid models

Building on the insights gained from traditional decision tree-based ensemble learning models, our research then explored the implementation and performance comparison of deep learning models. Deep learning, known for its ability to handle large and complex datasets, can offer promising improvements over traditional machine learning models, particularly in feature extraction and recognition of patterns that were not readily apparent. In this section, we conducted a comparative analysis between the best-performing decision tree-based ensemble learning models and their deep learning counterparts. The goal of this comparison was to determine whether the added complexity and computational requirements of deep learning models translate into significant performance gains in real-world applications such as energy management.

In an effort to advance power consumption prediction, this study systematically explored state-of-the-art deep learning, including LSTM, Bi-LSTM, GRU, Bi-GRU, one-dimensional CNN (1D-CNN), temporal convolutional network (TCN), and their hybrid variants such as LSTM-TCN, Bi-LSTM-TCN, GRU-TCN, and Bi-GRU-TCN. We extended these models with self-attention mechanisms to capture long-range dependencies and complex patterns in electricity consumption data [55]. Here, we considered Bi-GRU-TCN, also known as BiGTA-Net, which incorporates Bi-GRU, TCN, and the attention mechanism from the previous study, as

our benchmark model because we used the same input variables and data set (i.e., AEP) to demonstrate the superiority of this model [30]. Following our previous approach, we also included LSTM-TCN, Bi-LSTM-TCN, and GRU-TCN. Previous research has demonstrated that BiGTA-Net outperforms other state-of-the-art models, including LightGBM-sequence-to-sequence-attention-based Bi-LSTM (LGBM-S2S-Att-Bi-LSTM) [56], ranger-based online learning approach (RABOLA) [22], residual CNN with LSTM (ResCNN-LSTM) [57], and attention-based CNN with GRU (Att-CNN-GRU) [58].

To ensure robust evaluation, we standardized the following hyperparameters across experiments: the adaptive moment estimation (Adam) optimizer for its effectiveness with sparse gradients, a learning rate of 0.001 for steady convergence, and an attention mechanism to focus on significant temporal features for prediction accuracy. We used the leaky rectified linear unit (LeakyReLU) activation function because prior modeling showed that it outperformed the rectified linear unit (ReLU) and the scaled exponential linear unit [30], mitigating the dying ReLU problem while maintaining a small gradient when less active. A random state of 42 ensured the reproducibility of the result. The batch size was set to 24 to match the daily cycle of power consumption, and we ran 100 epochs to balance adequate learning and prevent overfitting. Following our previous research [30], we used a many-to-many multistep-ahead forecasting approach to predict 24 time points simultaneously, with the last time point (24th hour) serving as the day-ahead STLF for comparison with ensemble learning models.

To clarify, GBM achieved the best performance with the lowest MAPE (5.13), CVRMSE (7.67), and NMAE (5.35). As shown in Table 13, the stability and predictability of the dataset allowed deep learning models such as TCN (MAPE: 6.36, CVRMSE: 8.19, NMAE: 6.37) and Bi-GRU (MAPE: 6.41, CVRMSE: 8.51, NMAE: 6.54) to perform well. However, GBM outperformed them by efficiently exploiting the structured and less volatile nature of the data. It is important to note that the precision of the Bi-GRU-TCN may be suboptimal due to the fact that the values reflect only the 24th time point, rather than the average over all time points.

CatBoost demonstrated superior performance relative to other models, with a CVRMSE of 68.69 and NMAE of 42.61, as detailed in Table 14. Despite having a higher MAPE (41.35) than some deep learning models, CatBoost outperformed them in this regard. The high variability and presence of outliers in this dataset posed a significant challenge to the deep learning models, resulting in higher MAPE, CVRMSE, and NMAE values. For instance, the TCN exhibited a MAPE of 39.02, a CVRMSE of 90.09, and an NMAE of 51.00. Deep learning models, such as

**Table 13. Performance comparison between deep learning models and the GBM model on the University Residential Complex dataset.** All evaluation metrics are presented in percentage (%).

| Model | MAPE | CVRMSE | NMAE |
|---|---|---|---|
| Long short-term memory | 15.06 | 18.28 | 14.48 |
| Bidirectional long short-term memory | 7.87 | 10.06 | 8.00 |
| Gated recurrent unit | 8.54 | 11.52 | 8.62 |
| Bidirectional gated recurrent unit | 6.41 | 8.51 | 6.54 |
| One-dimensional convolutional neural network | 6.56 | 8.74 | 6.73 |
| Temporal convolutional network | 6.36 | 8.19 | 6.37 |
| Long short-term memory with temporal convolutional network | 6.63 | 8.73 | 6.78 |
| Bidirectional long short-term memory with temporal convolutional network | 7.14 | 9.23 | 7.18 |
| Gated recurrent unit with temporal convolutional network | 7.01 | 9.13 | 7.17 |
| Bidirectional gated recurrent unit with temporal convolutional network | 6.92 | 9.16 | 7.14 |
| Gradient boosting machine | 5.13 | 7.67 | 5.35 |

**Table 14. Performance comparison between deep learning models and the CatBoost model on the Appliances Energy Prediction dataset.** All evaluation metrics are presented in percentage (%).

| Model | MAPE | CVRMSE | NMAE |
|---|---|---|---|
| Long short-term memory | 38.50 | 88.84 | 50.11 |
| Bidirectional long short-term memory | 37.61 | 87.31 | 48.66 |
| Gated recurrent unit | 45.88 | 98.15 | 58.70 |
| Bidirectional gated recurrent unit | 48.17 | 92.46 | 55.78 |
| One-dimensional convolutional neural network | 39.95 | 87.84 | 49.66 |
| Temporal convolutional network | 39.02 | 90.09 | 51.00 |
| Long short-term memory with temporal convolutional network | 50.39 | 96.75 | 55.36 |
| Bidirectional long short-term memory with temporal convolutional network | 40.97 | 81.42 | 46.81 |
| Gated recurrent unit with temporal convolutional network | 44.13 | 89.13 | 52.77 |
| Bidirectional gated recurrent unit with temporal convolutional network | 34.31 | 86.19 | 46.58 |
| Categorical boosting | 41.35 | 68.69 | 42.61 |

Bi-LSTM and Bi-GRU, exhibited greater difficulty with this dataset, as evidenced by their higher MAPE (37.61), CVRMSE (87.31), and NMAE (48.66) values. The respective values were 61 and 48.17 for CVRMSE, 87.31 and 92.46 for NMAE, and 48.66 and 55.78 for MAPE. This can be attributed to the higher data requirements and sensitivity to noise exhibited by the models. The models exhibited higher values for CVRMSE (87.31 and 92.46, respectively) and NMAE (48.66 and 55.78, respectively), which can be attributed to the higher data requirements and sensitivity to noise exhibited by the models.

While it is prudent to be cautious in claiming superiority over deep learning models, the comparison between ensemble learning and deep learning models across these datasets illustrates the strengths of ensemble methods in handling diverse and noisy data with less extensive preprocessing and feature engineering. Their robustness, efficiency, and interpretability make them well-suited for electricity consumption forecasting, especially in datasets with high variability and shorter durations. This study underscores the importance of selecting an appropriate modeling approach that aligns with the characteristics of the data to achieve optimal forecasting performance.

## SHAP analysis of the optimal decision tree-based ensemble model

In Fig 4, we examine the applicability of SHAP by initially comparing the feature importance derived from the best-performing GBM on the University Residential Complex dataset with SHAP's summary plot. This comparison allows us to provide a foundational explanation of feature importance and to highlight the distinctions between these two methods.

Feature importance in the GBM model is a key metric that quantifies the impact of each feature on the predictive accuracy of the model. It is typically calculated based on the frequency and impact of each feature when used to partition the data in the model's decision trees. Fig 4 (a) clearly shows that features such as *Cons_avg* and *Cons_1* are heavily relied upon for predictions, underscoring their critical role in the model's decision-making process related to energy consumption.

In contrast, the SHAP summary plot, shown in Fig 4(b), provides a more detailed perspective. Specifically, it shows the impact of individual feature values on the model's output. For example, the feature *Holi* shows clear high and low SHAP values for its categories (0 and 1), reflecting significant differences in energy consumption on holidays versus nonholidays. In

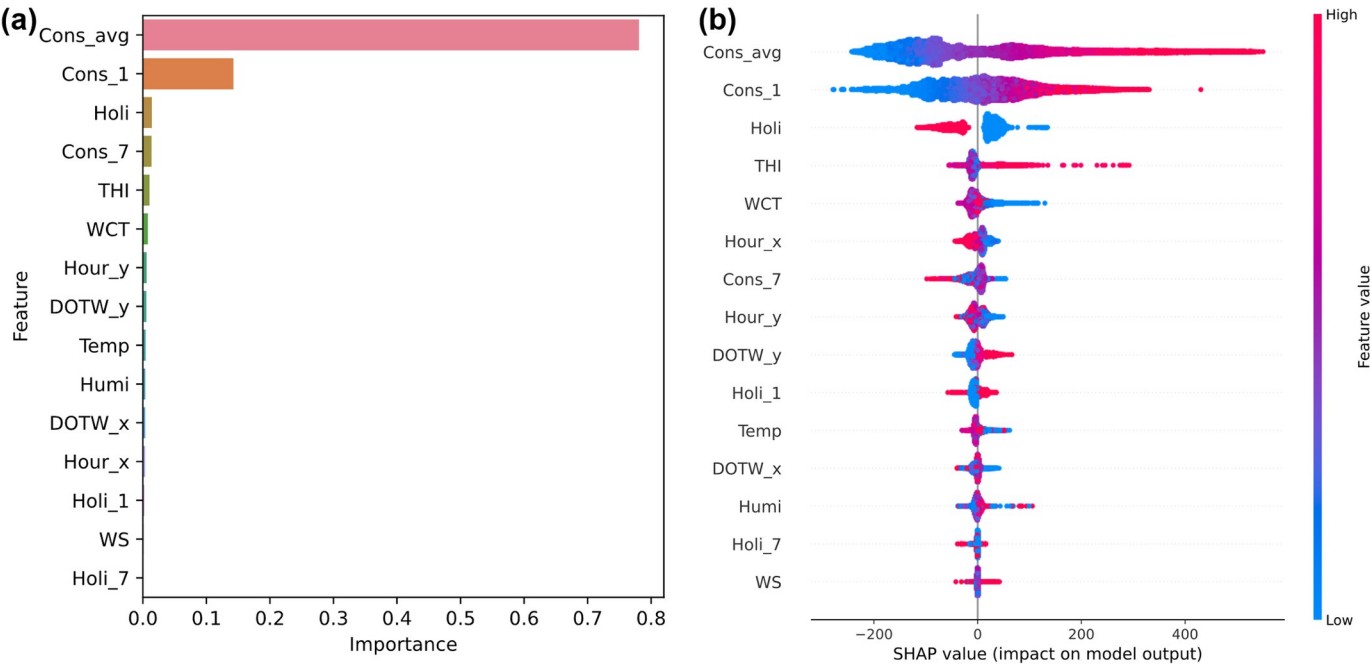

**Fig 4. Influence of input variables in the gradient boosting machine (GBM) model on the University Residential Complex dataset: (a) feature importance; (b) Shapley additive explanations (SHAP) summary plot.** See Table 2 for abbreviation definitions.

addition, features such as *THI* and *WCT* have SHAP values that vary significantly with high and low values, indicating the influence of seasonal conditions on energy consumption.

The superior depth of analysis provided by SHAP over traditional feature importance is obvious. SHAP not only details the contribution of each feature across different scenarios, but also enhances interpretability, allowing for a more comprehensive understanding of how specific conditions and features influence energy consumption. This distinction is critical to accurately interpreting model behavior and is thoroughly detailed in Fig 4 to ensure clarity and avoid potential misunderstandings regarding the application of these analysis methods.

Fig 5 shows SHAP summary plots for several ensemble learning models applied to the University Residential Complex dataset, including RF, XGBoost, LightGBM, and CatBoost. These plots provide a detailed comparison of how different models weigh the importance of different features and their impact on model outputs. For the RF model, in addition to *Cons_avg* and *Cons_1*, feature *Cons_7* also emerges as significantly influential, indicating that the model takes into account more diverse consumption metrics. Notably, for features such as *Holi* and *THI*, there is a tendency for higher values to lead to stronger negative and positive SHAP values, respectively, suggesting a direct correlation between these features and significant shifts in energy consumption patterns.

Both XGBoost and LightGBM show similarities to the GBM model, prioritizing *Cons_avg*, *Cons_1*, and *Holi* in terms of feature importance. This alignment suggests that these models share a common perspective on what most influences energy consumption within the dataset. Conversely, CatBoost shows a unique pattern where *Cons_1* is identified as the most important feature, followed by *Cons_avg*. There is also a wide distribution of SHAP values across other temporal features, which likely reflects CatBoost's ability to effectively handle categorical and time series data, demonstrating its learning capabilities. As seen in Table 11, GBM slightly

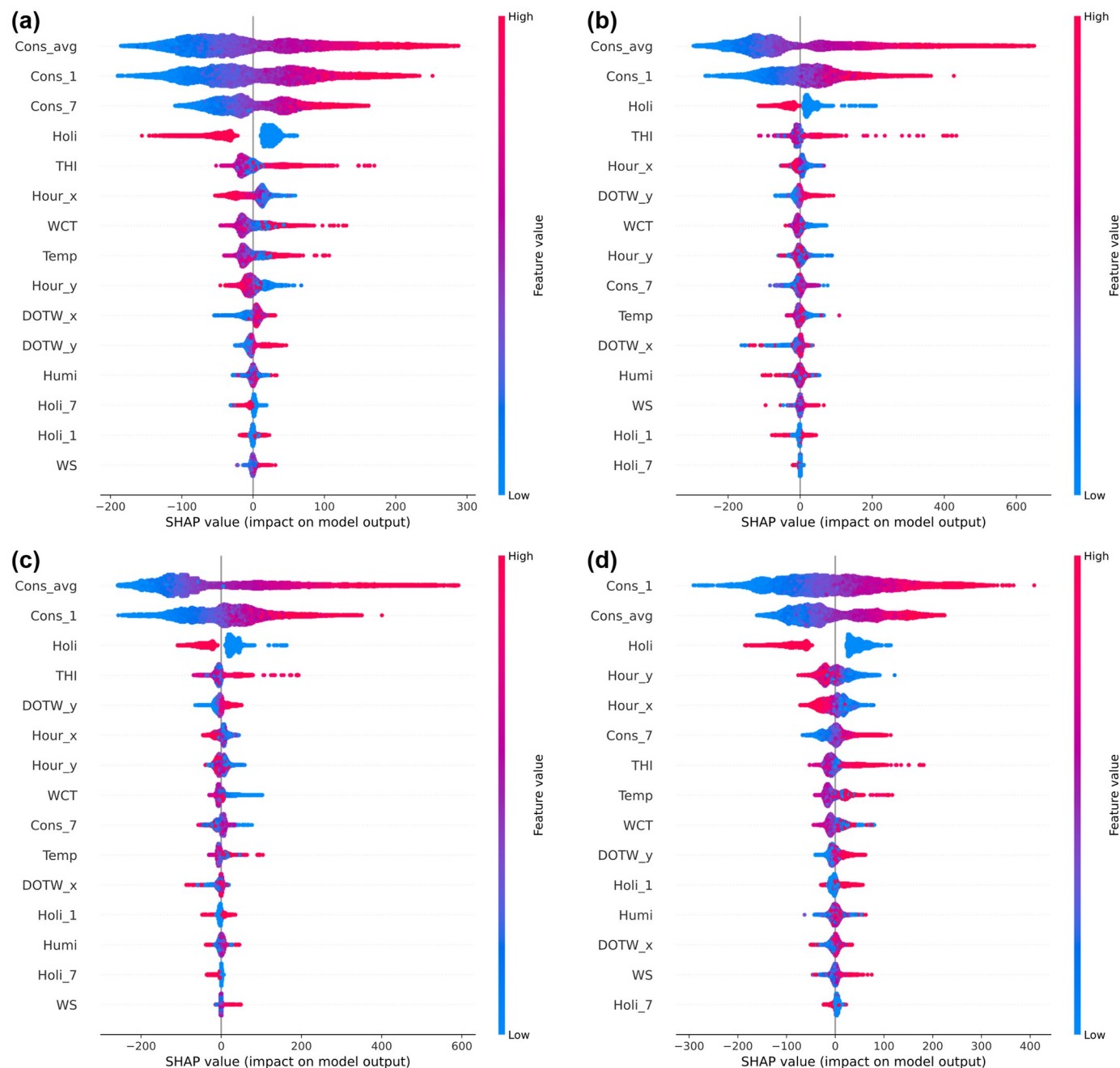

**Fig 5. SHAP summary plots of other ensemble learning models on the University Residential Complex dataset: (a) random forest (RF); (b) extreme gradient boosting (XGBoost); (c) light gradient boosting machine (LightGBM); (d) categorical boosting (CatBoost).** See Table 2 for abbreviation definitions.

outperforms the other models with an overall score of 5.86, effectively integrating key features such as *Cons_avg* and *Cons_1*.

Fig 6 illustrates how energy consumption (*Cons_avg*) in a university dormitory varies with different levels of *Holi*, where *Holi* = 0 is likely to be weekdays and *Holi* = 1 is holidays or weekends. Fig 6(a) shows a gradient of energy consumption from low to high. At lower *Cons_avg* levels (around 800 to 1000), energy consumption is minimal, likely reflecting early morning or

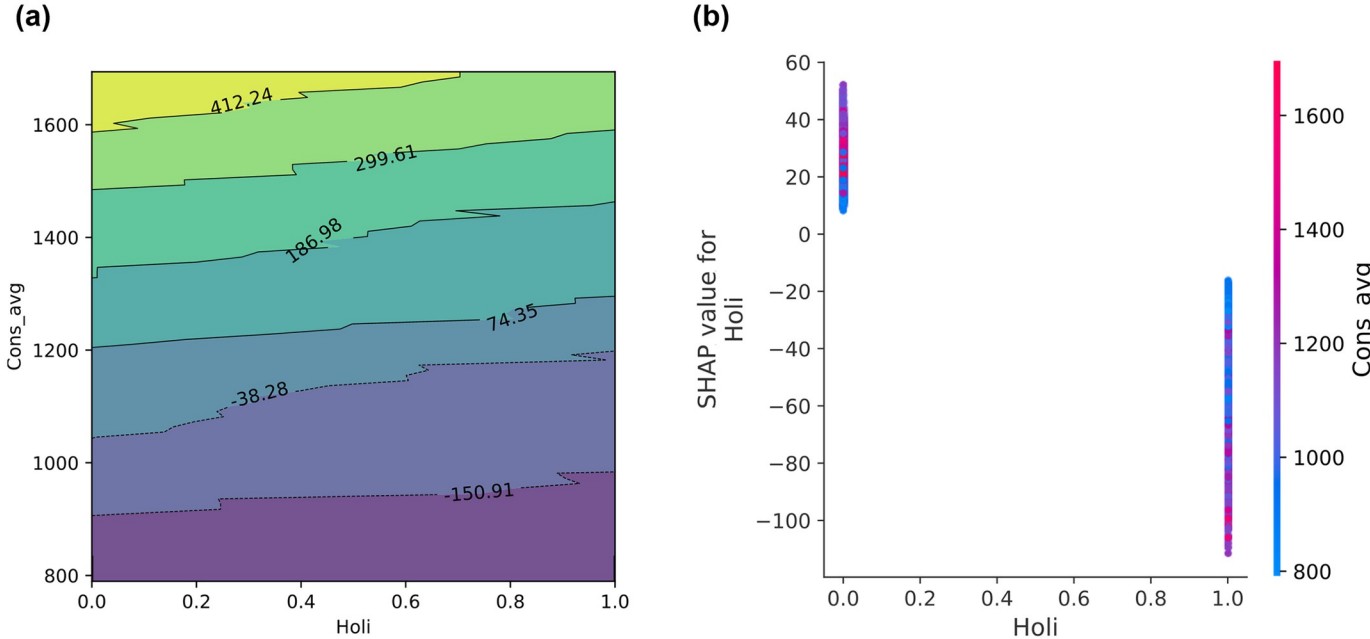

**Fig 6. Comparison of partial dependence plots (PDPs) from the GBM model on the University Residential Complex dataset: Analyzing the relationship between holiday status (*Holi*) and average electricity consumption (*Cons_avg*) (a) using scikit-learn; (b) using SHAP analysis.** See Table 2 for abbreviation definitions.

nonacademic hours. As *Cons_avg* increases (around 1200 to 1600), the plot shows that energy consumption peaks during holidays or weekends, highlighted by contours that rise to +412.24. Contrary to typical expectations of lower energy consumption on nonacademic days due to less operational activity, the plot indicates increased consumption on holidays. This suggests that residential activities such as heating, cooling, and appliance use are more prevalent when students are more likely to be indoors. This concise analysis highlights the unique energy consumption patterns in student housing, driven by the interplay of academic schedules and residential lifestyles that can lead to increased energy demand on noninstructional days.

The SHAP summary plot in Fig 6(b) clearly delineates the impact of the *Holi* variable on *Cons_avg*, providing an illustrative contrast to the more general results from the scikit-learn PDP. This plot clearly shows the binary impact of *Holi* on energy consumption predictions, with a marked difference between weekday and holiday SHAP values on weekdays (*Holi* = 0). The SHAP values for weekdays are predominantly positive, indicating that energy consumption predictions are higher on these days. This suggests that during regular academic days, when classes and university activities are in full swing, there is a significant demand for energy. In contrast, SHAP values on holidays (*Holi* = 1) are generally negative, indicating a decrease in energy consumption. This indicates that on days when the university complex is less active, such as during breaks or holidays, energy demand decreases significantly. This decrease could be due to reduced heating, cooling, and lighting needs when fewer people are on campus.

This accurate representation of SHAP values helps to understand the specific impact of weekdays and holidays on energy consumption within a university campus. By providing clarity on how different days contribute to energy consumption, this analysis can aid in more efficient energy management and planning, ensuring that energy resources are optimally utilized according to actual activity patterns within the complex.

**(a)** **(b)**

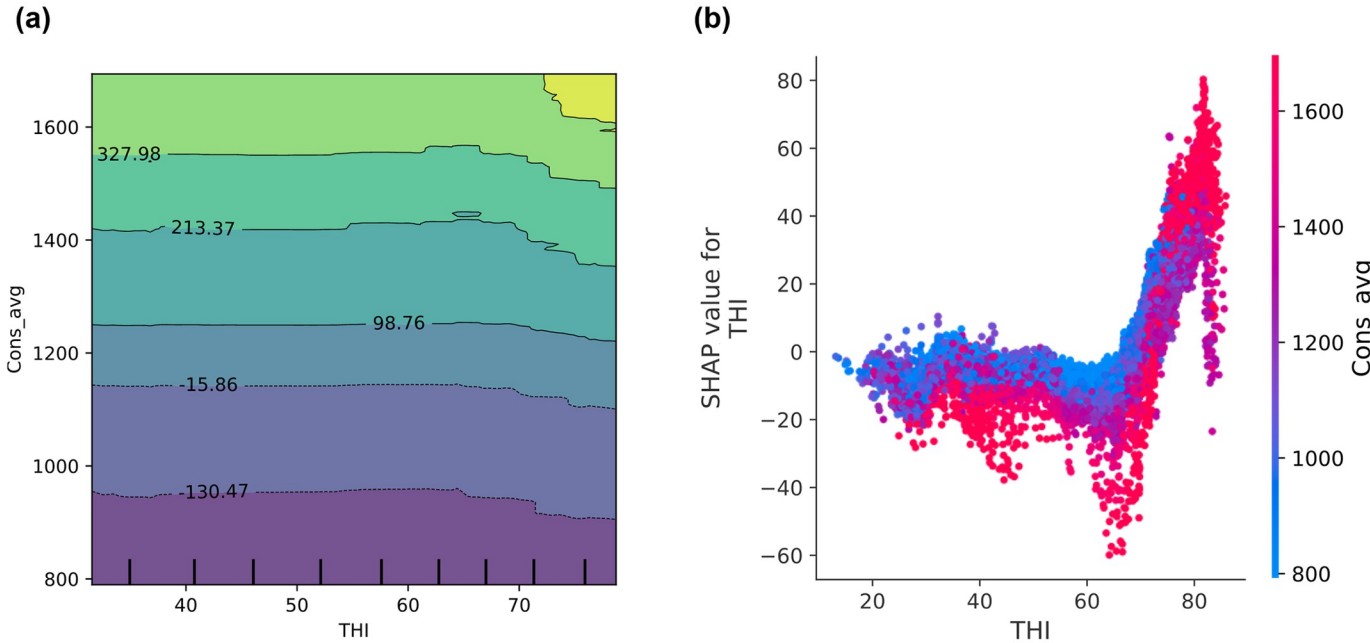

**Fig 7. Comparison of PDPs from the GBM model on the University Residential Complex dataset: Analyzing the relationship between temperature-humidity index (*THI*) and *Cons_avg* (a) using scikit-learn; (b) using SHAP analysis.** See Table 2 for abbreviation definitions.

Fig 7 provide contrasting analytical views of how the *THI* relates to *Cons_avg* in a university residential complex. The PDP for Fig 7(a) on the *THI* and *Cons_avg* reveals the impact of *THI* on energy consumption in a university residential complex. The plot demonstrates that decreased energy usage occurs at lower THI (20–40). This suggests that energy needs are lower under cooler, less humid conditions. A slight increase in energy consumption is observed at moderate THI (40–60). This indicates that minimal climate control is sufficient to maintain comfort at these levels. A significant increase in energy usage is observed at higher THI (60–80). This is driven by the need for extensive air conditioning to manage high heat and humidity.

The SHAP plot, Fig 7(b), presents individual data points and demonstrates how SHAP values (which indicate the impact on the model output) vary with THI. The plot reveals a high degree of variability in SHAP values, particularly as THI increases. Notably, between THI values of 20 to about 40, SHAP values are predominantly negative, aligning with the PDP's suggestion of reduced energy use. A pronounced increase in SHAP values is observed as THI continues to rise beyond 60. SHAP values shift markedly towards positive values, reaching a peak around THI 80. This sharp increase aligns with the PDP findings and suggests that high THI values are strong predictors of increased energy consumption. This may be attributed to the necessity of cooling systems to counteract uncomfortable or unsafe humidity and temperature levels.

While both scikit-learn's PDP and SHAP analysis provide insights into how THI influences energy consumption in a university residential complex, they differ significantly in detail and applicability. The PDP offers a broad view, showing general trends but potentially obscuring specific effects due to its averaged approach. In contrast, SHAP provides a detailed, granular look at the direct impacts of individual THI levels, offering clearer insights into how minor variations affect energy usage. This detailed perspective renders SHAP particularly valuable for

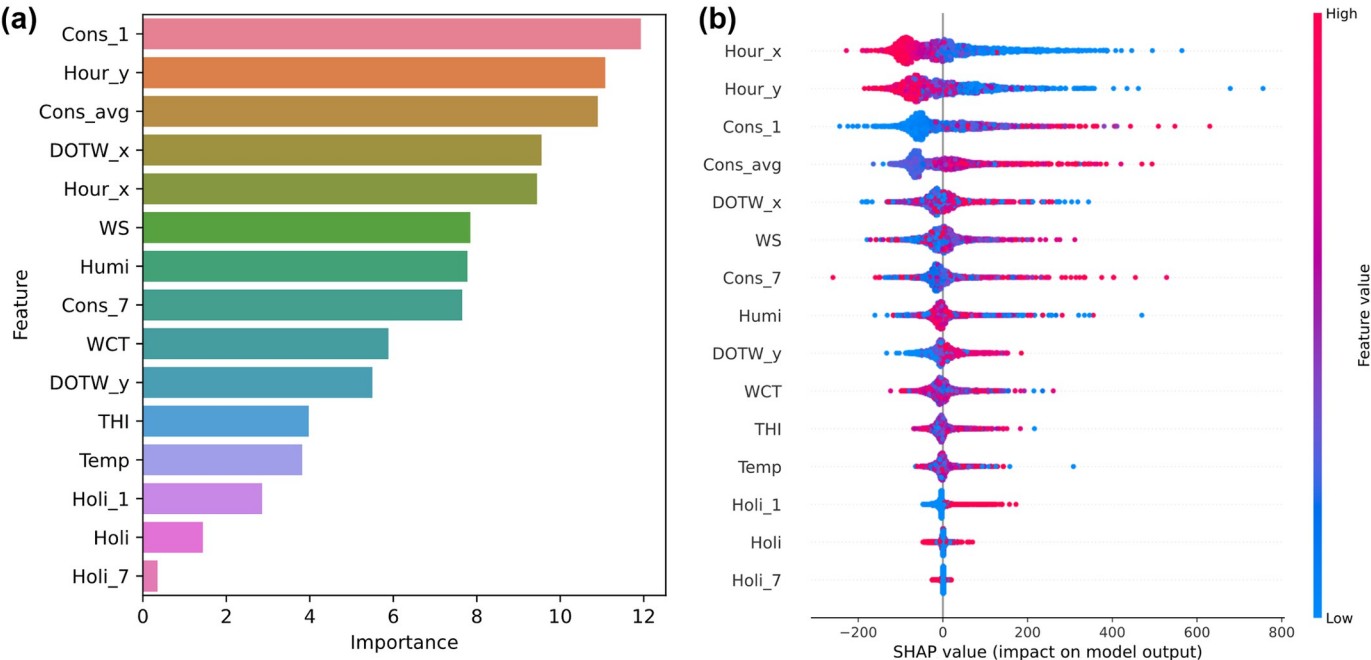

**Fig 8. Influence of input variables in the CatBoost model on the Appliances Energy Prediction dataset: (a) feature importance; (b) SHAP summary plot.** See Table 2 for abbreviation definitions.

precise energy management and predictive modeling, as it is capable of capturing the nuances necessary for effective decision-making in residential settings.

The feature importance metrics for the CatBoost model, shown in Fig 8(a), reveal that *Cons_1* and *Hour_y* are critical to the model's predictions, underscoring their significant role in shaping energy consumption patterns. Furthermore, considering that the dataset spans from January to April, *WCT* emerges as a more influential variable compared to *THI* in both figures. This prominence of *WCT* is pertinent during the colder months of the dataset, when wind chill is a critical determinant of electricity consumption, more so than humidity and temperature. Adapting the model to these seasonal variations improves its accuracy by effectively tailoring its predictions to the specific climatic conditions that prevail during these months.

The SHAP summary plot shown in Fig 8(b) and the feature importance plot both show a broad distribution of significance across variables, in contrast to previous analyses where specific variables typically stood out. This even distribution indicates that no single variable dramatically outweighs others in terms of influence on model results within the limited time frame of the dataset, which emphasizes time-related variables over weather-related factors such as *THI* and *WCT*. However, the SHAP summary plot reveals nuances that are not apparent in the feature importance plot. For example, *Hour_x* and *Hour_y* tend to have lower SHAP values as their values increase, suggesting that these time variables may be inversely related to energy consumption predictions. Conversely, *Cons_1* and *Cons_avg* have higher SHAP values as their values increase, indicating a direct and strong positive influence on the predicted energy consumption, a detail not explicitly revealed by the feature importance metrics.

In addition, the ranking of variables differs between the SHAP summary and feature importance plots. This discrepancy can be attributed to how each method measures influence: Feature importance considers the frequency and depth of a feature's use in the model's decision trees, which primarily reflects how the data are partitioned. In contrast, SHAP values represent

the actual impact of each feature value on the model's output, providing a direct view of its contribution to prediction shifts. This methodological difference underscores why SHAP can provide a more detailed perspective on feature contributions, especially in complex models where interactions and nonlinear relationships are significant. These findings highlight the complementary nature of using both feature importance and SHAP analysis in model interpretation, providing a more complete understanding of how features affect model predictions under different conditions.

In Fig 9, SHAP summary plots for different ensemble learning models applied to the Appliances Energy Prediction dataset—specifically RF, GBM, XGBoost, and LightGBM—show a distinct pattern of feature importance compared to CatBoost. In particular, internal factors such as *Cons_avg* and *Cons_1* are significantly more influential in these models than in CatBoost, where the time variables *Hour_x* and *Hour_y* are emphasized. This difference is likely due to CatBoost's ability to handle categorical variables, which in this context are divided into 24 categories, allowing for more nuanced learning of temporal patterns. As a result, temporal patterns appear to improve prediction accuracy more effectively than historical energy consumption metrics.

Furthermore, similar to the observations in the University Residential Complex dataset, the shortened duration of the dataset leads to a reduced significance of weather-related variables in all models, including CatBoost. In addition, the significance of holiday information is generally low, suggesting that differences in electricity use between weekdays and weekends are minimal. This may indicate the prevalence of remote working practices during the dataset period, which is consistent with actual circumstances [9]. This nuanced understanding helps to clarify the subtle differences in the impact of features across models, highlighting the adaptability of CatBoost in learning significant time-based patterns that potentially improve predictive performance.

Fig 10(a) presents a PDP that elegantly maps the distribution of SHAP values and their interactions between temporal variables, specifically *Hour_x* and *Hour_y*. This figure decodes time using a novel approach, where the values for *Hour_x* are encoded to represent 6 p.m. (denoted as −1) and 6 a.m. (denoted as 1), and *Hour_y* values correspond to noon (indicated by −1) and midnight (indicated by 1). This figure illustrates that the increasing SHAP values for *Hour_y* positively influence the CatBoost model construction around noon, whereas decreasing values have a negative effect around midnight. Based on the pivotal moments of 6 a.m. (in red) and 6 p.m. (in blue), where the SHAP value for *Hour_y* is neutral, the SHAP value trends positively from dawn to noon and inversely from noon to dusk in the left segment of the figure. Likewise, the right segment reveals a positive SHAP value trajectory from dusk to midnight and a negative one from midnight until dawn. Fig 10(b) is a PDP offering a view of the SHAP value distribution and interaction with the holiday indicator in relation to *Hour_y*. The SHAP value distribution skews more negatively on holidays in comparison to weekdays, although the magnitude of this discrepancy is not as pronounced as for other features. This observation implies that consistent home office activity may occur even on holidays, and the referenced literature [9] supports this finding.

Fig 11 offers a clear interpretation of the GBM model's predictions for the test set of the Appliances Energy Prediction dataset. While the accuracy of this model is not significantly superior to that of the University Residential Complex dataset, the decision plot provides a detailed understanding of the model's decision-making process, highlighting its advantages over the LIME method. The decision plot illustrates the influence of various features on the model's output values. The plot reveals that key features, such as *Hour_x*, *Hour_y*, and *Cons_1*, exhibit significant divergences, indicating their substantial influence on the model's

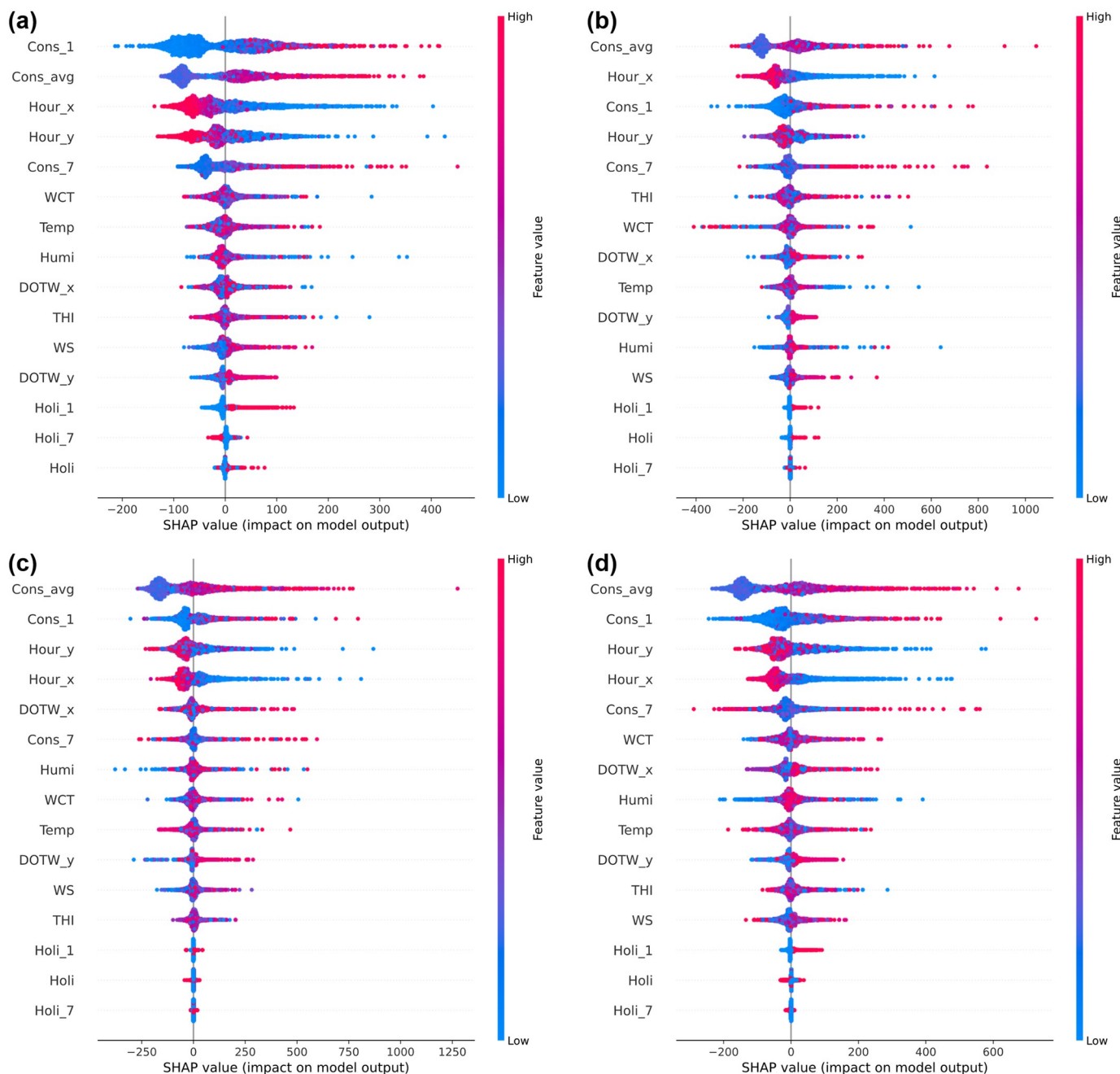

**Fig 9. SHAP summary plots of other ensemble learning models on the Appliances Energy Prediction dataset: (a) RF; (b) GBM; (c) XGBoost; (d) LightGBM.**
See Table 2 for abbreviation definitions.

predictions. This visualization enables us to discern how each feature contributes to the prediction, with a color gradient from blue to red indicating the magnitude of the model output values.

In comparison to LIME, the decision plot offers several advantages. It provides a comprehensive overview of the model's behavior across the entire test set, rather than merely local explanations for individual predictions. This global perspective is essential for comprehending

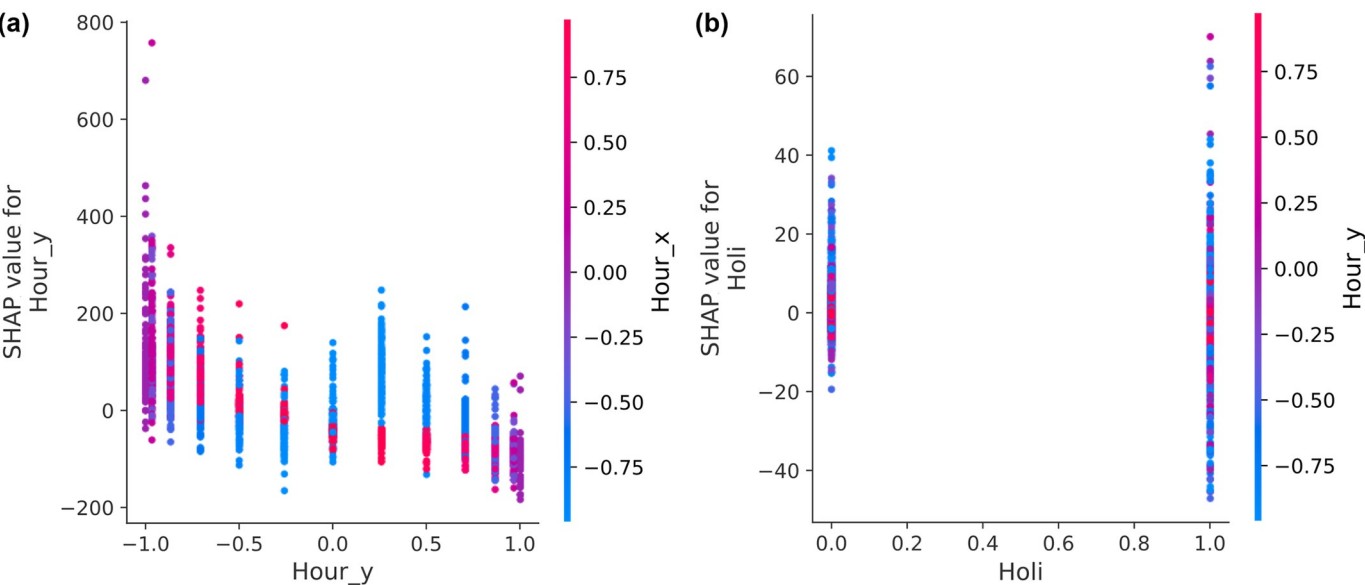

**Fig 10. PDPs of the CatBoost model on the Appliances Energy Prediction dataset: (a)** *Hour_x* **and** *Hour_y***; (b) holiday indicator (***Holi***) and** *Hour_y***.** See Table 2 for abbreviation definitions.

the overall behavior of the model and identifying trends. Additionally, the decision plot visualizes the interactions between multiple features, revealing intricate relationships that may not be evident with LIME. Moreover, the decision plot is also computationally efficient, rendering it practical for use with large datasets. In contrast to LIME, which may yield inconsistent explanations due to its reliance on perturbing input data, the decision plot provides consistent insights that reflect the model's inherent decision path for each prediction.

To scrutinize the correlation between electricity consumption and smart sensor readings, we analyzed the AEP dataset, sampled hourly. A detailed enumeration of the input variables from the AEP dataset is readily accessible in the referenced literature [9]. To identify the quintessential predictive model, we constructed and trained five decision tree-based ensemble learning models fine-tuned with optimal hyperparameter values. These models were employed to forecast hourly electricity consumption. Fig 12 indicates that the RF model achieved the most commendable $R^2$ value among its counterparts.

Thus, Fig 13 exhibits a comprehensive summary plot for the entire AEP dataset using the RF model. Within this figure, the variable pertaining to lights (representing light energy consumption) emerged as the most influential, whereas the week status (indicating a weekend (0) or weekday (1)) was the least consequential regarding model predictive power. Additionally, a closer examination revealed that relative humidity (*RH*)-related variables (i.e., *RH_1*, *RH_out*, *RH_8*, and *RH_3*) exerted a substantial effect on the RF model architecture. This insight underscores the significance of atmospheric moisture content in shaping the model's understanding of electricity consumption patterns.

## Discussion

The study examined the performance of ensemble models, including RF, GBM, LightGBM, and CatBoost, in structured data scenarios and demonstrated their superiority over deep learning models. This observation was supported by similar conclusions from Bellahsen and Dagdougui [17] and Zhang et al. [18]. Specifically, CatBoost was able to significantly reduce

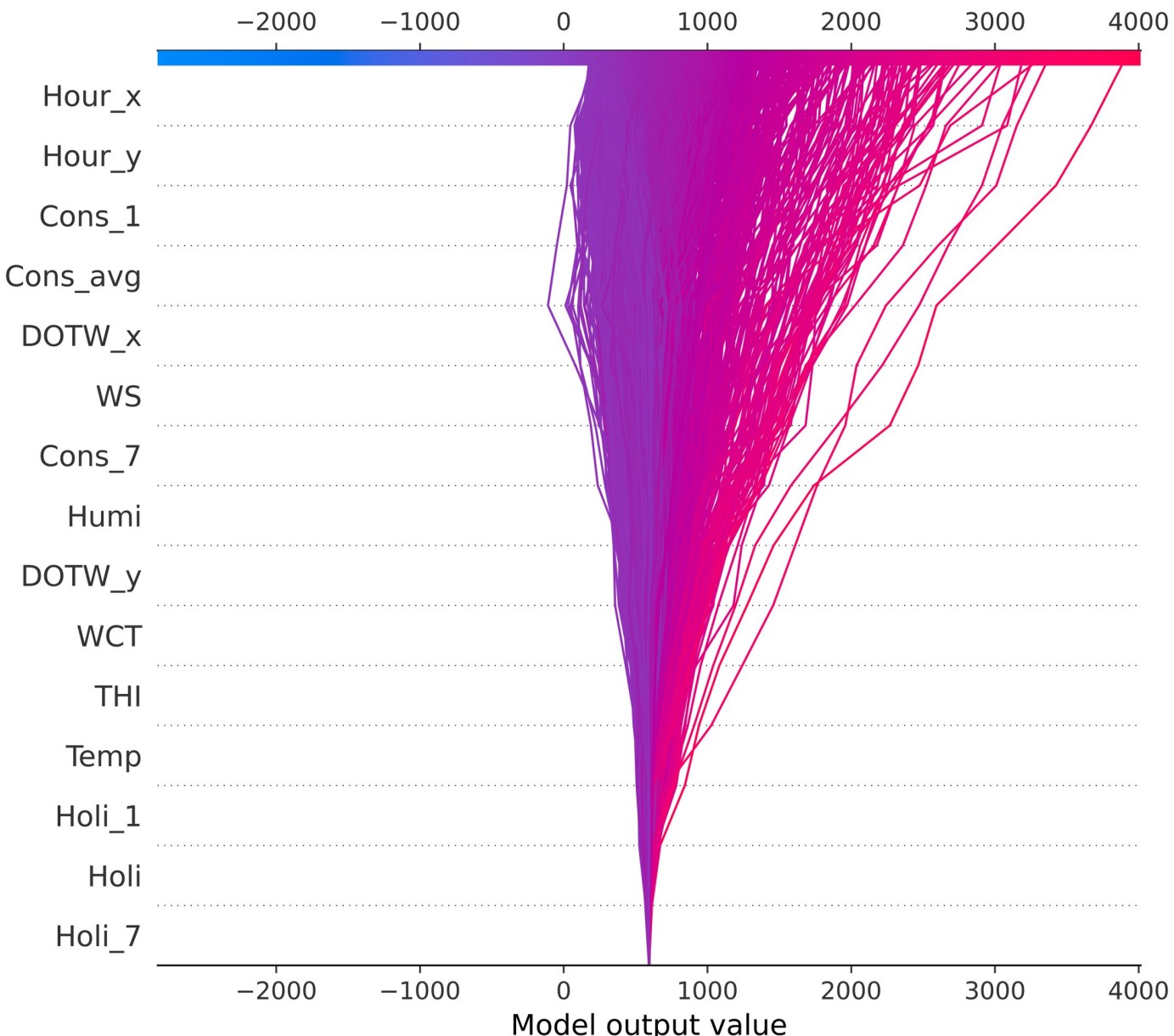

**Fig 11. SHAP decision plot for the test set on the Appliances Energy Prediction dataset.** See Table 2 for abbreviation definitions.

CVRMSE by effectively learning from timestamp information categorized in smaller datasets. The resilience of these models to overfitting and their efficient data handling, especially in the presence of noisy data points, highlighted their suitability for energy consumption forecasting. To bridge the performance gap across scales, strategies included applying ensemble learning models to residential data in various environments, comparing multiple ensemble learning models with deep learning alternatives, and using diverse XAI techniques beyond SHAP.

- Ensemble learning methods such as GBM and CatBoost demonstrated superior performance in environments characterized by high variability and the presence of outliers, which are common in datasets such as the Appliances Energy Prediction dataset. These models utilized

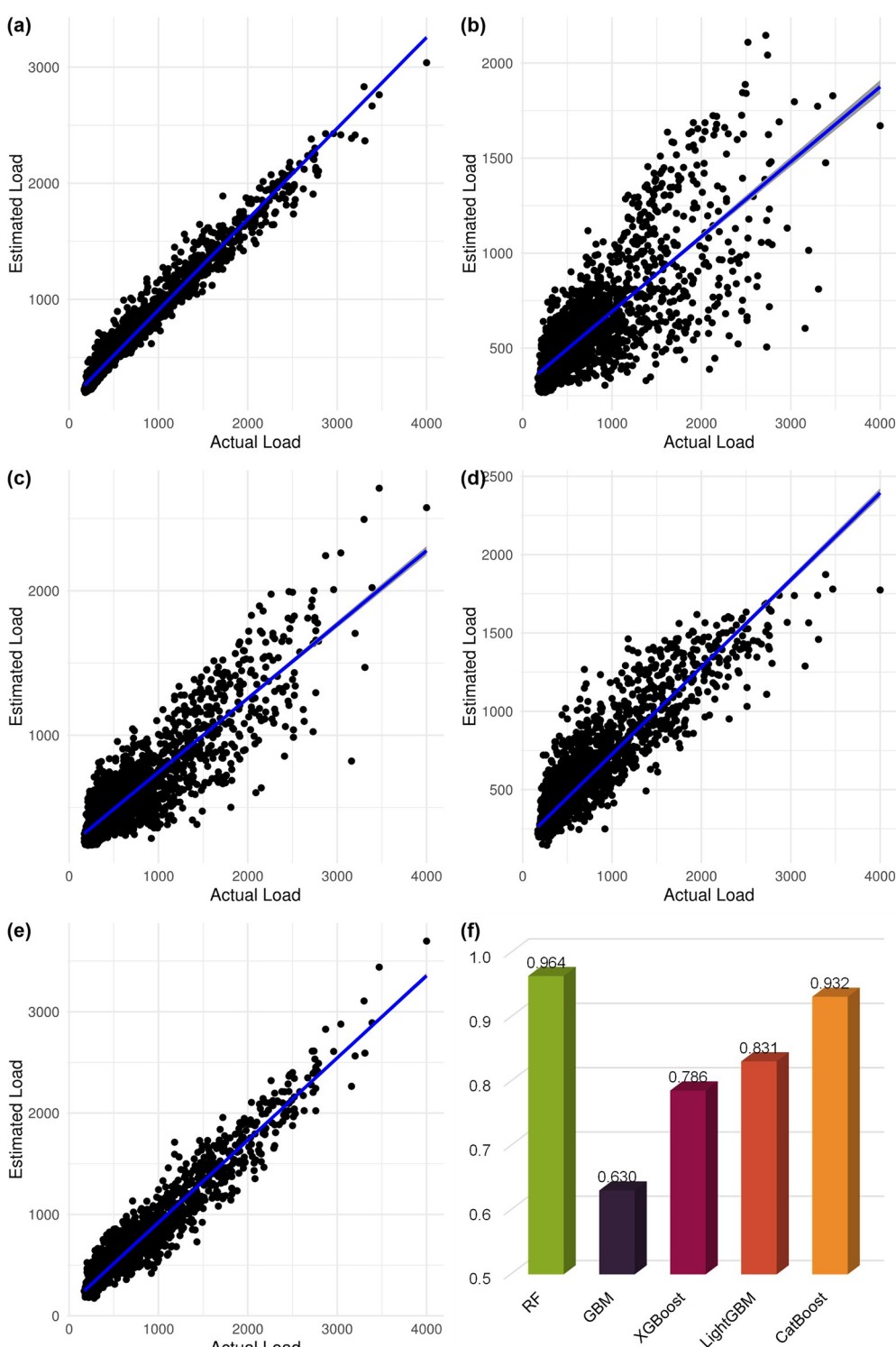

**Fig 12. Scatterplots and $R^2$ comparison for each decision tree–ensemble learning model.** (a) random forest; (b) gradient boosting machine; (c) extreme gradient boosting; (d) light gradient boosting machine; (e) categorical boosting; (f) $R^2$ comparison.

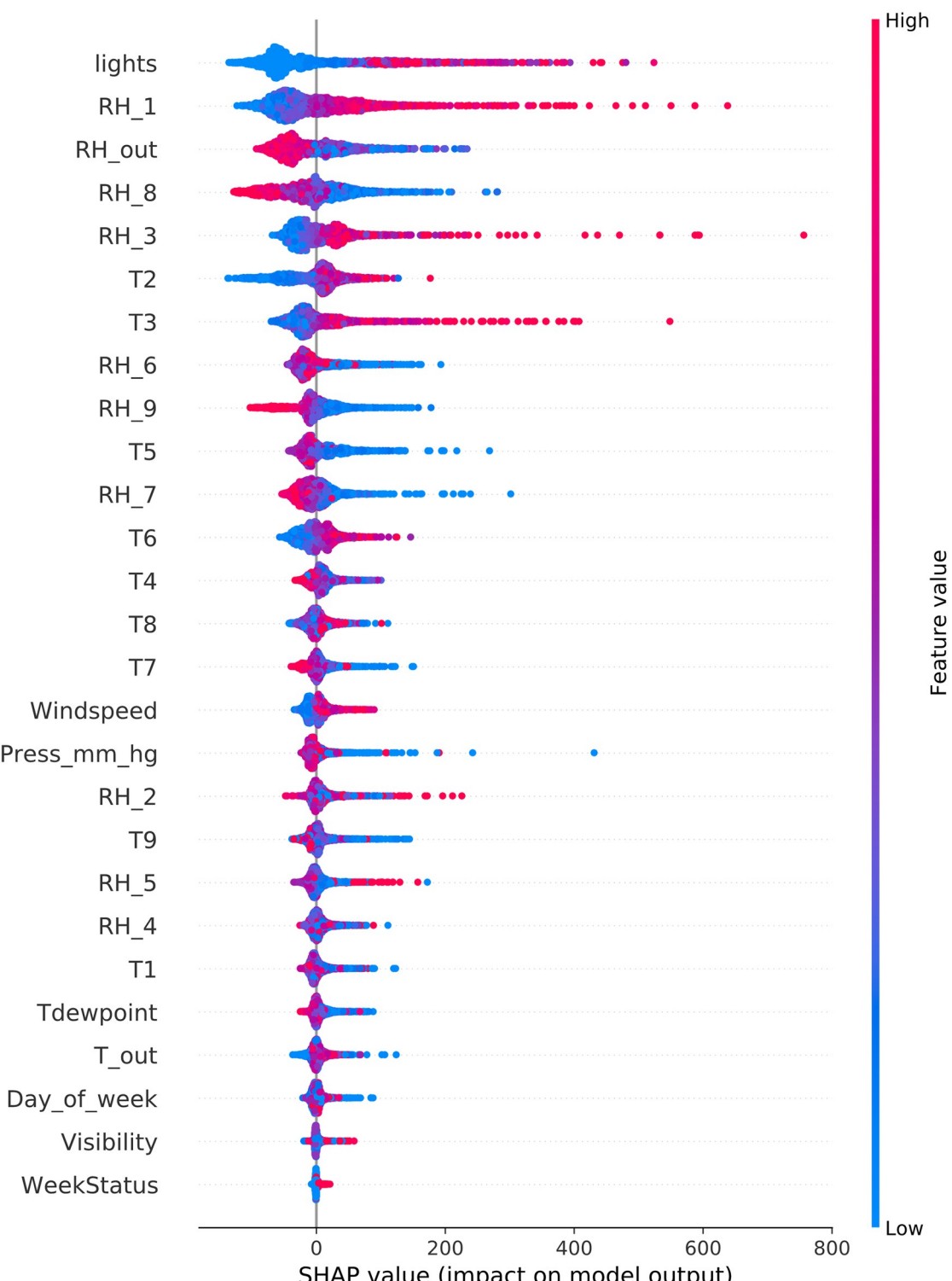

**Fig 13. Summary plot of the RF model on the hourly Appliances Energy Prediction raw dataset.**

multiple decision trees and employed techniques such as bagging and boosting to reduce the impact of noisy data points, thus maintaining high accuracy and reliability under varied conditions.

- In contrast to deep learning models, which often necessitated extensive preprocessing to effectively handle diverse input features, ensemble learning models automatically selected and weighted features based on their importance. This capability was particularly beneficial for handling complex datasets with cyclical time variables and binary indicators, which were crucial for accurate electricity consumption forecasting.

- Ensemble learning models generally required less data to achieve high performance and provided more interpretable outcomes compared to deep learning models. This was particularly advantageous in datasets with limited size or shorter duration, such as the Appliances Energy Prediction dataset, where ensemble methods could converge more rapidly and offer more transparent insights into feature contributions.

In conclusion, the refined discussion highlighted why ensemble learning models were sometimes more effective than deep learning models in handling the complexities of energy usage contexts. By focusing on tailored solutions and enhancing data-driven strategies, the predictive accuracy for both large-scale and small-scale energy prediction scenarios could be significantly improved. This analysis contributed to the academic discourse and provided a strategic direction for future research, ensuring our models were not only effective but also comprehensible and applicable across various domains.

The practical application of the Tree SHAP library with these models provided deeper interpretive insights, suggesting that decision tree-based models coupled with SHAP could effectively reveal complex interactions within the data. The integration of Tree SHAP has yielded promising results in domains beyond our own, including finance and healthcare. This is evidenced by the findings of Jabeur et al. [59] and Safaei et al. [60]. Future studies could investigate the applicability of these models and Tree SHAP in diverse domains using a range of datasets to further validate and enhance the generalizability of our findings.

The study employed SHAP analysis to elucidate the significance of variables such as average consumption ($Cons\_avg$) and consumption in the previous hour ($Cons\_1$). These findings corroborate the pivotal role of historical consumption patterns highlighted by Moon et al. [32] and Lahouar and Slama [61]. Furthermore, our findings corroborate the influence of external factors, such as holidays and the temperature-humidity index, on energy consumption, in accordance with the observations of Li et al. [62]. This comparative analysis not only validated the use of SHAP but also expanded our understanding by contrasting the contributions of these variables to energy prediction models. Notably, while our results were consistent with previous studies in many respects, they also introduced nuanced differences in the magnitude of the effects of these variables. This discrepancy may be attributed to the specifics of the dataset or the model configurations employed, which could be clarified by further investigation.

Although the results were encouraging, they were not without limitations. The high variability and presence of outliers in datasets such as the Appliances Energy Prediction dataset posed significant challenges. These factors required the use of advanced preprocessing techniques to construct realistic input variables, effectively deal with data leakage, and ensure the reliability of the models. In addition, the computational demands associated with deep learning models and even some ensemble learning methods highlighted the need for more efficient computational strategies, especially for real-time applications. Future research should aim to extend the time frame of the dataset and include more diverse environmental and operational

variables. This could help to capture broader trends and further test the adaptability of the models under different conditions.

The integration of XAI, particularly SHAP, into complex models has proven to be a significant challenge, with issues ranging from the handling of high-dimensional data to meeting computational demands. These challenges have underscored the need for advances in computational efficiency and algorithm optimization to make XAI more practical and accessible. Additionally, as deep learning continues to evolve, there is a critical need to improve the explainability of these models. The application of advanced XAI techniques could have made even the most sophisticated deep learning architectures more interpretable and trustworthy [63–65]. The refined discussion illuminated the intricate roles of various predictor variables and the relative efficacy of ensemble and deep learning models. It underscored the necessity for sustained enhancements in model development, XAI integration, and algorithm optimization to effectively address the complexities of energy consumption forecasting. This analysis not only contributed to the academic discourse but also provided a strategic direction for future research. The process ensured that our models were not only effective but also understandable and applicable across domains. This was achieved by clarifying the ongoing evolution and potential of XAI.

To bridge the gap in model performance across different scales, it will be imperative to enhance our strategies around data collection and model customization.

- The quantity and quality of data collected from smaller units can be enhanced to provide models with a richer basis for learning and adaptation. Our study will demonstrate the superior effectiveness of ensemble models such as RF, LightGBM, CatBoost, GBM, and XGBoost over deep neural networks in structured data scenarios by examining their performance on diverse datasets. The resilience of these models to overfitting and their efficient handling of noisy data points will highlight their suitability for various forecasting scenarios.

- The development of specialized models or the customization of existing ones to better suit the granularity of smaller datasets will result in more accurate predictions. The practical application of the Tree SHAP library with these models will provide deeper interpretive insights, as decision tree-based models coupled with SHAP are effective in revealing complex interactions within the data. Furthermore, SHAP will be employed to identify and eliminate irrelevant features or to combine meaningful variables, thereby enhancing the integrity of our models. Additionally, SHAP will be utilized to verify the accuracy of the models generated, ensuring their relevance and effectiveness for practical applications.

- The incorporation of advanced analytics, which entails the utilization of sophisticated machine learning techniques and analytical tools such as feature engineering and deep learning, has the potential to enhance the adaptability and accuracy of models in a multitude of application settings. While our findings are encouraging, they underscore the necessity for continued enhancements in model development, XAI integration, and algorithm optimization to effectively address the complexities of energy consumption forecasting.

In order to optimize the application of XAI within the energy sector, particularly in complex systems such as combined heat and power (CHP) systems and networked microgrids (NMGs), it is of the utmost importance to harness predictive analytics in a way that extends beyond traditional modelling. Although the foundational studies of [66, 67] do not directly incorporate advanced machine learning, their focus on optimizing energy consumption and managing intricate power flows demonstrates the potential for integrating XAI. The application of XAI to enhance electricity usage forecasts enables the provision of actionable insights, both for operational optimization and strategic planning within CHP and NMG setups. The

predictive power of XAI can transform raw data into understandable and actionable information, thereby facilitating operational adjustments that optimize energy production and distribution based on real-time demand and supply conditions.

Furthermore, the implementation of XAI can facilitate the integration of complex algorithmic resolutions and practical applications, thereby enabling the execution of sophisticated analyses that are accessible and interpretable to non-experts in the field. This transparency is of paramount importance for the purpose of gaining stakeholder trust and for the wider adoption of AI-driven tools in traditional energy sectors. In essence, the integration of XAI into energy prediction models specifically tailored for systems such as CHP and NMGs can significantly enhance their efficiency and reliability. By providing a clear understanding of how and why decisions are made, XAI not only improves the usability of predictive models but also ensures that these models can be effectively applied in real-world scenarios, driving forward the evolution of energy management practices. This strategic enhancement of XAI applications promises to expand its impact, making it a pivotal tool in the advancement of energy management technologies.

## Conclusions

The pursuit of energy efficiency within household buildings is a cornerstone in the advancement of smart home technology. The widespread adoption of the AEP dataset for residential energy efficiency underscores its significance within the research community. In this investigation, we applied the proposed methodology to the residential structure using the AEP dataset. We integrated external and internal factors to shape the input variables, mitigating data leakage concerns. Subsequently, we partitioned the dataset into distinct training and testing sets and designed five decision tree-based ensemble learning models on the training set. These models were rigorously evaluated against MAPE, CVRMSE, and NMAE metrics using the same testing set.

The findings reveal that the GBM model, honed with only external factors, outperformed its counterparts in forecasting energy efficiency at the household building level within the AEP dataset. The SHAP analysis illuminated that THI and WCT wielded a more profound influence on the model architecture than the conventional weather metrics of temperature, humidity, and wind speed. Furthermore, timestamp information emerged as a more dominant force in the GBM model compared to meteorological data. The PDP analysis for the holiday indicator implied the prevalence of telecommuting, providing home energy management system administrators with actionable insight for behavioral pattern estimation and efficient planning.

Future research endeavors will investigate the applicability of the proposed methodology to other types of buildings, including commercial facilities and industrial complexes. This expansion will permit an assessment of its adaptability and effectiveness across diverse environmental and usage contexts, providing insights into energy consumption patterns that differ significantly from those observed in residential settings. By testing the methodology in these varied settings, we aim to refine and adapt our models to capture the distinctive characteristics of each building type effectively. Furthermore, we will integrate data on renewable energy sources, such as solar irradiance and photovoltaic output, with occupancy patterns into our forecasting models. This integration will not only enhance the precision of the models but also increase their relevance by reflecting real-world, dynamic energy generation and usage scenarios. Incorporating these diverse data streams will enable us to develop more sophisticated, context-aware models that are capable of predicting energy needs with greater accuracy. The incorporation of occupancy patterns, in particular, will facilitate the generation of more

detailed energy demand forecasts that align with actual building usage patterns, thereby enabling more efficient energy management and planning.

To address potential challenges such as data heterogeneity and privacy concerns in these diverse settings, we will develop robust data harmonization and privacy-preserving methods. Our efforts will also focus on innovating an online dynamic learning-based energy forecasting model to address the challenge of data scarcity and augment energy efficiency at the building level. We will aim to develop advanced energy forecasting models that encompass multistep-ahead and probabilistic frameworks, designing an architecture to explain the inner workings of these models. Ultimately, our objective is to develop an electrical energy management and trading system that is grounded in robust demand and supply forecasting, which incorporates external factors such as consumer behavior. This system will facilitate energy, environmental, and economic management. The integration of XAI into this system, particularly in the context of complex setups such as CHP and NMGs, has the potential to significantly enhance not only the predictive accuracy of our models but also their transparency and applicability. This integration could be pivotal in advancing energy management technologies. We anticipate that these initiatives will provide a comprehensive foundation for future advancements in energy management and contribute significantly to the field.

## Author Contributions

**Conceptualization:** Jihoon Moon.

**Data curation:** Jihoon Moon.

**Formal analysis:** Jihoon Moon, Muazzam Maqsood.

**Funding acquisition:** Yunyoung Nam.

**Investigation:** Muazzam Maqsood, Sung Wook Baik, Seungmin Rho, Yunyoung Nam.

**Methodology:** Jihoon Moon, Muazzam Maqsood, Dayeong So, Sung Wook Baik.

**Project administration:** Seungmin Rho.

**Software:** Dayeong So, Seungmin Rho.

**Supervision:** Muazzam Maqsood, Sung Wook Baik, Seungmin Rho, Yunyoung Nam.

**Validation:** Muazzam Maqsood, Yunyoung Nam.

**Visualization:** Seungmin Rho.

**Writing – original draft:** Jihoon Moon.

**Writing – review & editing:** Yunyoung Nam.

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
