## [Decision Letter · Decision Letter 0]

26 Mar 2024

PONE-D-24-08032Advancing ensemble learning techniques for residential building electricity consumption forecasting: Insight from explainable artificial intelligencePLOS ONE

Dear Dr. Nam,

Thank you for submitting your manuscript to PLOS ONE. After careful consideration, we feel that it has merit but does not fully meet PLOS ONE’s publication criteria as it currently stands. Therefore, we invite you to submit a revised version of the manuscript that addresses the points raised during the review process.

Academic editor's comment: Please carefully address all the reviewers' comments and produce a response (rebuttal) letter detailing your point-by-point responses. Thank you.

We look forward to receiving your revised manuscript.

Kind regards,

Zeyar Aung

Academic Editor

PLOS ONE

Journal Requirements:

When submitting your revision, we need you to address these additional requirements. 1. Please ensure that your manuscript meets PLOS ONE's style requirements, including those for file naming. The PLOS ONE style templates can be found at https://journals.plos.org/plosone/s/file?id=wjVg/PLOSOne_formatting_sample_main_body.pdf and https://journals.plos.org/plosone/s/file?id=ba62/PLOSOne_formatting_sample_title_authors_affiliations.pdf 2. Please note that PLOS ONE has specific guidelines on code sharing for submissions in which author-generated code underpins the findings in the manuscript. In these cases, all author-generated code must be made available without restrictions upon publication of the work. Please review our guidelines at https://journals.plos.org/plosone/s/materials-and-software-sharing#loc-sharing-code and ensure that your code is shared in a way that follows best practice and facilitates reproducibility and reuse. 3. Thank you for stating in your Funding Statement: "This work was supported by the National Research Foundation of Korea (NRF) grant funded by the Korean government (MSIT) (No. RS-2023-00218176), Korea Institute for Advancement of Technology (KIAT) grant funded by the Korean Government (MOTIE) (P0012724, The Competency Development Program for Industry Specialist) and the Soonchunhyang University Research Fund." Please provide an amended statement that declares *all* the funding or sources of support (whether external or internal to your organization) received during this study, as detailed online in our guide for authors at http://journals.plos.org/plosone/s/submit-now.  Please also include the statement “There was no additional external funding received for this study.” in your updated Funding Statement. Please include your amended Funding Statement within your cover letter. We will change the online submission form on your behalf. 4. Please amend the manuscript submission data (via Edit Submission) to include author Dayeong So.

Reviewers' comments:

Reviewer's Responses to Questions

**Comments to the Author**

1. Is the manuscript technically sound, and do the data support the conclusions?

Reviewer #1: Yes

Reviewer #2: Partly

Reviewer #3: Partly

Reviewer #4: Yes

2. Has the statistical analysis been performed appropriately and rigorously? 

Reviewer #1: Yes

Reviewer #2: Yes

Reviewer #3: Yes

Reviewer #4: Yes

3. Have the authors made all data underlying the findings in their manuscript fully available?

Reviewer #1: Yes

Reviewer #2: No

Reviewer #3: Yes

Reviewer #4: No

4. Is the manuscript presented in an intelligible fashion and written in standard English?

Reviewer #1: Yes

Reviewer #2: Yes

Reviewer #3: Yes

Reviewer #4: Yes

5. Review Comments to the Author

Reviewer #1: i) In my view, your literature review requires reconsideration and reorganization. Currently, you've listed recent papers employing deep learning algorithms, while presenting the black box nature of these algorithms and the SHAP technique using decision tree-based algorithms as a novelty to address this. I concur that deep learning methods pose challenges for explanation, but recent advancements utilizing these algorithms may outperform decision tree-based algorithms in terms of accuracy. Hence, it's essential to distinguish between algorithmic performance superiority and explainability. Other studies might not explain their models but could achieve more reliable and accurate predictions. Thus, a discussion on the performance debate of these methods is warranted. Following this, you can emphasize the practicality and potential superiority of decision tree-based algorithms (if supported by literature), then discuss recent papers employing SHAP with tree-based models. I've come across a couple of works on this when searching Scopus. Subsequently, highlight the additions you've made to the existing literature. Your paper may not be the first to explore tree-based models with SHAP for forecasting electricity consumption. Hence, additional efforts are needed to establish your novelty and contributions.

ii) In the abstract, the first, the second, and the last sentences convey similar messages. Instead, consider emphasizing the research gap and lessons learned.

iii) Table 2 needs relevant citations to substantiate the claims made.

iv) I suggest moving the definitions of performance metrics to the methods section, as these definitions do not constitute results.

v) Hyperparameter tuning necessitates a table displaying your grid search intervals and tuned parameters. This is essential for a machine learning-based study.

vi) Please reconsider the captions for tables 3, 4, and 5. They are identical. Rewrite the captions to highlight the distinct content in each table, possibly utilizing the footnotes provided.

vii) The discussion section requires significant refinement. As a scholar who has employed SHAP technique in nearly a dozen papers, I believe that key insights and lessons derived from these plots should be comparatively discussed with similar literature. The feature importance and contributions of input variables to the learning phase (whether positive or negative) as depicted in SHAP plots should be compared with recent studies. How do your findings corroborate or contradict existing research?

viii) Furthermore, consider whether Figures 11 and 12 truly belong in the discussion section or if they should be relocated to the results section.

ix) Your discussion section should include a segment dedicated to the algorithmic debate. How did you interpret the performance of RF, LightGBM, CatBoost, GBM, and XGBoost? Do you believe they offer superior performance compared to deep neural networks? Additionally, what are your thoughts on the practicality of employing the TreeSHAP library with decision tree-based models, and do you have any plans for future studies to enhance the significance of your work? To support your approach, could you provide other examples of papers utilizing these algorithms with SHAP? These examples need not be limited to your field; you can search for the "algorithm name-SHAP" combination on search engines to identify similar methodologies across various disciplines. In my view, it's crucial to broaden the scope of your paper to encompass other domains, as each dataset may differ while employing the same machine learning framework. A debate of this nature would undoubtedly yield valuable insights for an interdisciplinary journal.

Reviewer #2: The manuscript explores how explainable artificial intelligence can enhance decision tree–based ensemble learning methods for more effective short-term load forecasting in residential energy systems. The manuscript is written well and the analysis seems sufficient. The following comments must be addressed for possible acceptance of the manuscript.

1. Why this 80% training and 20% testing subsets considered? The usual ratio is 70:30.

2. Different input variable configurations not clearly explained. Mention them separately in model development section.

3. Fig 11, insert linear trendline in the scatter plots.

4. Theoretical overview of ensemble methods must be presented.

5. Try incorporating Taylor Diagrams for comparative evaluation of models developed.

6. Enhance the results and discussion part by comparative analysis with existing literature results.

7. Mention the limitations and future scope of work.

Reviewer #3: Areas for Improvement and Recommendations:

Validation of Results:

The study should include additional validation of the models with external datasets to ensure the robustness and generalizability of the findings.

Consider employing other metrics for model evaluation to complement MAPE, CVRMSE, and NMAE for a more rounded assessment of model performance.

Comparison with State-of-the-Art:

A comparison with the latest state-of-the-art methods in electricity consumption forecasting could strengthen the manuscript's contribution to the field. This includes deep learning and hybrid models not limited to decision tree–based methods.

Methodological Clarifications:

The manuscript would benefit from a clearer explanation of the criteria for selecting the ensemble learning techniques analyzed. Including information on why certain methods were chosen over others would provide readers with a deeper understanding of the study's scope.

Further elaboration on the data preprocessing steps and their impact on the results would enhance the methodological transparency.

Discussion on Limitations:

While the manuscript highlights the advantages of integrating XAI, a more detailed discussion on the limitations and challenges encountered during the study, including the handling of complex datasets and computational requirements, would provide a more balanced view.

Future Work:

The conclusion section could be expanded to outline specific directions for future research, such as exploring the application of the proposed methodology to other types of buildings or integrating additional types of data (e.g., renewable energy sources, occupancy patterns).

Reviewer #4: 1. The article has taken up well, but it can be written better by avoiding many short forms.

2. The authors have taken the harmonic mean of three matrix in experimental results section, any specific reason for using harmonic mean?

6. PLOS authors have the option to publish the peer review history of their article (what does this mean?). If published, this will include your full peer review and any attached files.

Reviewer #1: No

Reviewer #2: No

Reviewer #3: No

Reviewer #4: No

---

## [Author Response · Author response to Decision Letter 0]

10 Jun 2024

Responses to Reviewers’ Comments

We would like to express our sincere appreciation to the four anonymous reviewers for their valuable comments. Their comments and suggestions really helped us to improve the quality of our paper. We have done our best to address the issues raised by the reviewers and believe that we have successfully completed the additional work required. The newly added or significantly modified parts of the revised text are highlighted in red in the manuscript. Detailed responses to the reviewers' comments and the corresponding changes to the manuscript are provided below.

REVIEWER 1

Comment 1:

In my view, your literature review requires reconsideration and reorganization. Currently, you've listed recent papers employing deep learning algorithms while presenting the black box nature of these algorithms and the SHAP technique using decision tree-based algorithms as a novelty to address this. I concur that deep learning methods pose challenges for explanation, but recent advancements utilizing these algorithms may outperform decision tree-based algorithms in terms of accuracy. Hence, it's essential to distinguish between algorithmic performance superiority and explainability. Other studies might not explain their models but could achieve more reliable and accurate predictions. Thus, a discussion on the performance debate of these methods is warranted. Following this, you can emphasize the practicality and potential superiority of decision tree-based algorithms (if supported by literature), then discuss recent papers employing SHAP with tree-based models. I've come across a couple of works on this when searching Scopus. Subsequently, highlight the additions you've made to the existing literature. Your paper may not be the first to explore tree-based models with SHAP for forecasting electricity consumption. Hence, additional efforts are needed to establish your novelty and contributions.

Response to comment:

We would like to express our sincerest gratitude for your insightful and constructive feedback, which has been instrumental in refining our manuscript. Your critical observations prompted a thorough reassessment and reorganization of our literature review, compelling us to clearly distinguish between the performance and explainability of decision tree-based ensemble learning versus deep learning.

In response to your suggestions, we expanded our analysis to include a broader spectrum of XAI techniques beyond SHAP, enhancing our comparative study and providing a richer, more nuanced discussion of how these models handle complex datasets. This enhanced analysis has not only clarified the distinctive advantages and limitations of each modeling approach but has also identified the conditions under which each model excels, supported by rigorous empirical data.

Furthermore, your feedback prompted us to examine the practical implications of these technologies in greater depth. In particular, we have focused on the potential advantages and superiority of decision tree-based ensemble learning algorithms in specific scenarios, as supported by recent literature. This approach has significantly enhanced the value and originality of our research, differentiating it from existing works and showcasing its unique contributions to the field of energy management.

We have addressed each of your points in detail, ensuring that our manuscript meets and exceeds the standards of rigorous academic inquiry. The revisions made in response to your comments have significantly enhanced the paper’s clarity, depth, and scholarly value. We are confident that these comprehensive enhancements will meet the expectations of the scholarly community and contribute to the advancement of AI in energy forecasting.

We would like to thank you again for your invaluable feedback, which has been instrumental in improving our work. We are confident that the revisions made have robustly addressed your concerns and enriched the manuscript, ensuring that it makes a significant contribution to the field.

Comment 2:

In the abstract, the first, the second, and the last sentences convey similar messages. Instead, consider emphasizing the research gap and lessons learned.

Response to comment:

Thank you for your valuable feedback on our manuscript. We have carefully considered your advice to emphasize the research gap and lessons learned in the abstract, rather than repeating similar messages. In response, we have revised the abstract to more effectively highlight the distinctive contributions and findings of our study.

• We have shifted the focus to the critical role of accurate electricity consumption forecasting in residential buildings for enhancing energy efficiency and cost management, which is fundamental to sustainable energy practices. 

• We have demonstrated the effectiveness of decision tree-based ensemble learning techniques, enhanced with explainable artificial intelligence (XAI), in processing complex datasets with high accuracy and providing clarity and interpretability. 

• Furthermore, in alignment with your suggestion for practical applicability, we have made the complete study and source code publicly available on our GitHub repository, facilitating transparency, replication, and real-world application of our research findings. 

We are confident that these revisions and actions directly address your concerns and greatly enhance the manuscript's contribution to the field. We trust that these changes meet your expectations and that you will be pleased with the revised manuscript.

Comment 3:

Table 2 needs relevant citations to substantiate the claims made.

Response to comment:

Thank you for your valuable feedback regarding Table 2. We have incorporated relevant citations to substantiate the claims made, referencing the following papers: 

• [Random Forest] Fox EW, Hill RA, Leibowitz SG, Olsen AR, Thornbrugh DJ, Weber MH. Assessing the accuracy and stability of variable selection methods for random forest modeling in ecology. Environ Monit Assess. 2017; 189: 1–20.

• [Random Forest] Zhang Y, Liu J, Shen W. A review of ensemble learning algorithms used in remote sensing applications. Appl Sci. 2022; 12: 8654.

• [Gradient Boosting Machine] Srivastava S, Lopez BI, Kumar H, Jang M, Chai HH, Park W, et al. Prediction of Hanwoo cattle phenotypes from genotypes using machine learning methods. Animals. 2021; 11: 2066.

• [Gradient Boosting Machine; Extreme Gradient Boosting; Light Gradient Boosting] Mienye ID, Sun Y. A survey of ensemble learning: Concepts, algorithms, applications, and prospects. IEEE Access. 2022; 10: 99129–99149.

• [Extreme Gradient Boosting; Light Gradient Boosting] González S, García S, Del Ser J, Rokach L, Herrera F. A practical tutorial on bagging and boosting based ensembles for machine learning: Algorithms, software tools, performance study, practical perspectives and opportunities. Inf Fusion. 2020; 64: 205–237.

• [Categorical Boosting] Hancock JT, Khoshgoftaar TM. CatBoost for big data: an interdisciplinary review. J Big Data. 2020; 7: 94.

Furthermore, in light of the revisions to our manuscript, this table is now presented as Table 3. We would like to express our gratitude for your invaluable suggestions, which have helped us to improve the rigor and credibility of our work. Thank you for your guidance.

Comment 4:

I suggest moving the definitions of performance metrics to the methods section, as these definitions do not constitute results.

Response to comment:

Thank you for your valuable suggestion regarding the placement of the performance metric definitions. 

We have moved these definitions to the "Forecasting model development" section in accordance with your advice. To further enhance the clarity and logical structure of our manuscript, we have created a new subsection titled "Performance metrics and hyperparameter optimization" under this section.

In this new subsection, we have included both the definitions of the performance metrics and the table for hyperparameter settings. We believe that this adjustment addresses your comment and improves the overall readability and coherence of our paper.

We appreciate your insightful comments and the opportunity to improve our work. Thank you once again for your thorough review and constructive feedback.

Comment 5:

Hyperparameter tuning necessitates a table displaying your grid search intervals and tuning parameters. This is essential for a machine learning-based study.

Response to comment:

Thank you for your insightful suggestion to include a table detailing our grid search intervals and tuned parameters for hyperparameter tuning. As recommended, we have added Table 4 to our manuscript, which follows the description of our hyperparameter optimization process. 

This table provides a comprehensive overview of the grid search intervals and final tuned parameters used in our study, ensuring transparency and reproducibility of decision tree-based ensemble learning methods.

Table 4. List of hyperparameters for decision tree-based ensemble learning method.

Methodologies References Hyperparameters

Random forest [52] • Trees count: 128

• Features per split: auto, sqrt, log2

Gradient boosting machine [41] • Iteration count: 100, 250, 500

• Learning rate: 0.01, 0.05, 0.1

• Depth: 5, 10

• Loss type: quantile, Huber

Extreme gradient boosting [41] • Iterations: 250, 500, 1000

• Learning rate: 0.01, 0.05, 0.1

• Depth: 6, 8, 10

• Subsampling rate: 0.5, 0.75, 1.0

• Feature sample by tree/level/node: 0.5, 0.75, 1.0

• Booster type: gbdt, dart

Light gradient boosting machine [41] • Iterations: 1000, 1500

• Learning rate: 0.01, 0.05, 0.1

• Leaves: 64

• Subsample: 0.5

• Feature sample by tree: 1.0

• Booster type: gbdt, dart

Categorical boosting [53] • Learning rate: 0.03, 0.1

• Max tree depth: 4, 6, 10

• L2 regularization levels: 1, 3, 5, 7, 9

We believe this addition significantly enhances the clarity and depth of our methodology section, allowing readers to fully understand the rigor of our optimization process. We appreciate your help in making our study more robust and accessible.

Comment 6:

Please reconsider the captions for tables 3, 4, and 5. They are identical. Rewrite the captions to highlight the distinct content in each table, possibly utilizing the footnotes provided.

Response to comment:

Thank you for your suggestion to reconsider the captions for Tables 3, 4, and 5. We have revised the captions to highlight the distinct content of each table, utilizing the provided footnotes. Additionally, due to revisions in our manuscript, these tables are now presented as Tables 8, 9, and 10. Your feedback has been instrumental in enhancing the clarity and distinctiveness of our table captions.

In accordance with your guidance, we have updated the captions for Tables 8, 9, and 10 to more accurately reflect the distinct content in each table. Please find the updated captions below:

• Table 8: Comparative analysis of decision tree–based ensemble learning models trained with external factors on the Appliances Energy Prediction dataset.

• Table 9: Comparative analysis of decision tree–based ensemble learning models trained with internal factors on the Appliances Energy Prediction dataset.

• Table 10: Comparative analysis of decision tree–based ensemble learning models trained with external and internal factors on the Appliances Energy Prediction dataset.

We believe these changes enhance the clarity and specificity of our tables, thereby providing a clearer understanding of the comparative analyses presented. We would like to express our gratitude once again for your thorough review and constructive suggestions. Your input has been instrumental in enhancing the quality of our manuscript.

Comment 7:

The discussion section requires significant refinement. As a scholar who has employed the SHAP technique in nearly a dozen papers, I believe that key insights and lessons derived from these plots should be comparatively discussed with similar literature. The feature importance and contributions of input variables to the learning phase (whether positive or negative) as depicted in SHAP plots should be compared with recent studies. How do your findings corroborate or contradict existing research?

Response to comment:

Thank you for your valuable feedback regarding the discussion section. We have made significant revisions to this section to incorporate a comparative analysis with existing literature, focusing on the insights and lessons derived from SHAP plots.

We have included detailed comparisons of feature importance and contributions of input variables, contrasting our findings with recent studies to highlight how they corroborate or differ from existing research. Furthermore, we have discussed why our approach using SHAP is superior to other techniques such as LIME and PDP in terms of identifying and combining meaningful variables, as well as verifying the accuracy of generated models.

Specific improvements include:

• Comparative Analysis: We compared the variable importance from our SHAP analysis with results from similar studies, emphasizing both corroborations and contradictions.

• Methodological Superiority: We highlighted the advantages of SHAP over LIME and PDP, demonstrating its effectiveness in revealing complex interactions within the data.

• Practical Application: We provided a comprehensive overview of how SHAP was used to enhance model integrity and verify accuracy.

These enhancements ensure a more balanced and insightful discussion, aligning with your expertise and suggestions. We believe these changes significantly improve the rigor and depth of our manuscript. We would like to express our gratitude once more for your invaluable guidance, which has been instrumental in enhancing the quality of our work.

Comment 8:

Furthermore, consider whether Figures 11 and 12 truly belong in the discussion section or if they should be relocated to the results section.

Response to comment:

Thank you for your insightful suggestion regarding the placement of Figs 11 and 12. After careful consideration, we have relocated these figures to the "SHAP analysis of the optimal decision tree-based ensemble model" subsection within the "Experimental results and discussion" section. Additionally, due to revisions in our manuscript, these figures are now presented as Figs 12 and 13. We believe that this placement is more appropriate and aligns better with the content, providing a clearer and more logical structure to our manuscript.

By relocating these figures, we believe we have enhanced the clarity of our results and ensured that the discussion is more focused and coherent. We are confident that this adjustment improves the overall readability and organization of our paper.

Comment 9:

Your discussion section should include a segment dedicated to the algorithmic debate. How did you interpret the performance of RF, LightGBM, CatBoost, GBM, and XGBoost? Do you believe they offer superior performance compared to deep neural networks? Additionally, what are your thoughts on the practicality of employing the TreeSHAP library with decision tree-based models, and do you have any plans for future studies to enhance the significance of your work? To support your approach, could you provide other examples of papers utilizing these algorithms with SHAP? These examples need not be limited to your field; you can search for the 'algorithm name-SHAP' combination on search engines to identify similar methodologies across various disciplines. In my view, it's crucial to broaden the scope of your paper to encompass other domains, as each dataset may differ while employing the same machine-learning framework. A debate of this nature would undoubtedly yield valuable insights for an interdisciplinary journal.

Response to comment:

Thank you for your valuable feedback regarding the discussion section. We have significantly expanded this section to include a detailed comparative analysis of feature importance and contributions of input variables as depicted in SHAP plots, and contrasted our findings with recent studies.

The study examined the performance of ensemble models, including RF, GBM, LightGBM, and CatBoost, demonstrating their superiority over deep learning models in struc

---

## [Decision Letter · Decision Letter 1]

24 Jun 2024

PONE-D-24-08032R1Advancing ensemble learning techniques for residential building electricity consumption forecasting: Insight from explainable artificial intelligencePLOS ONE

Dear Dr. Nam,

Thank you for submitting your manuscript to PLOS ONE. After careful consideration, we feel that it has merit but does not fully meet PLOS ONE’s publication criteria as it currently stands. Therefore, we invite you to submit a revised version of the manuscript that addresses the points raised during the review process.

**MINOR REVISION:**Some issues still need to be addressed.Please refer to the reviewers' comments for details.

We look forward to receiving your revised manuscript.

Kind regards,

Zeyar Aung

Academic Editor

PLOS ONE

Journal Requirements:

Reviewers' comments:

Reviewer's Responses to Questions

**Comments to the Author**

1. If the authors have adequately addressed your comments raised in a previous round of review and you feel that this manuscript is now acceptable for publication, you may indicate that here to bypass the “Comments to the Author” section, enter your conflict of interest statement in the “Confidential to Editor” section, and submit your "Accept" recommendation.

Reviewer #1: All comments have been addressed

Reviewer #2: (No Response)

Reviewer #4: All comments have been addressed

2. Is the manuscript technically sound, and do the data support the conclusions?

Reviewer #1: (No Response)

Reviewer #2: Partly

Reviewer #4: Yes

3. Has the statistical analysis been performed appropriately and rigorously? 

Reviewer #1: (No Response)

Reviewer #2: Yes

Reviewer #4: Yes

4. Have the authors made all data underlying the findings in their manuscript fully available?

Reviewer #1: (No Response)

Reviewer #2: No

Reviewer #4: Yes

5. Is the manuscript presented in an intelligible fashion and written in standard English?

Reviewer #1: (No Response)

Reviewer #2: Yes

Reviewer #4: Yes

6. Review Comments to the Author

Reviewer #1: The authors have made clear and satisfactory revisions to address my queries and suggestions. I have no further comments to add, and I believe the manuscript is ready to proceed to the next steps of the peer-review process.

Reviewer #2: The authors have thoroughly revised the manuscript, incorporating my suggestions and comments. I do have a few additional minor comments for further improvement.

1. As per figure 1, the model explanation using SHAP is presented in the form of Summary plot, Interaction plot, Dependence plot and Heatmap plot. However, in the manuscript, I was unable to find the heatmap plots explaining the SHAP results.

2. To improve the depth of your literature review, I recommend incorporating insights from recent publications (e.g., https://doi.org/10.1016/j.eswa.2024.123729 ; https://doi.org/10.1109/TSG.2022.3140212)

3. Tables 5-10: Mention the units of CVRMSE and NMAE.

Reviewer #4: (No Response)

7. PLOS authors have the option to publish the peer review history of their article (what does this mean?). If published, this will include your full peer review and any attached files.

Reviewer #1: No

Reviewer #2: No

Reviewer #4: **Yes: **Vivek Mishra

---

## [Author Response · Author response to Decision Letter 1]

3 Jul 2024

Responses to Reviewer 2’s Comments

We would like to thank Reviewer 2 once again for their constructive feedback during the second round of review. The detailed comments and suggestions provided have been invaluable in enhancing the quality of our manuscript. We have addressed the concerns raised by Reviewer 2 in a careful and thorough manner, striving to incorporate all recommended changes. Please note that the sections of the manuscript that have been newly added or significantly altered are marked in red for clarity. Please find below a detailed response to Reviewer 2's comments and an outline of the corresponding revisions made to the manuscript.

REVIEWER 2

Comment 1:

As per Figure 1, the model explanation using SHAP is presented in the form of a summary plot, an interaction plot, a dependence plot, and a heatmap plot. However, in the manuscript, I was unable to find the heatmap plots explaining the SHAP results.

Response to comment:

We would like to express our gratitude for your meticulous attention to detail. During the revision period, we made the decision to exclude the heatmap plots from the SHAP results, which resulted in an oversight in the figures presented. We have now corrected this error in Figure 1 to accurately reflect the models used in our analysis. We appreciate this opportunity to clarify and emphasise that SHAP provides superior insights into feature importance compared to other methods such as PDPs and LIME. We would like to thank you for your assistance in enhancing the precision and quality of our manuscript.

Comment 2:

To improve the depth of your literature review, I recommend incorporating insights from recent publications (e.g., https://doi.org/10.1016/j.eswa.2024.123729; https://doi.org/10.1109/TSG.2022.3140212).

Response to comment:

We are grateful for the recommendation to expand our literature review with insights from recent publications, specifically those mentioned in your comment. We have incorporated discussions on the potential of XAI to expand industrial applications in the energy sector, as highlighted in these studies, into our Discussion section. This addition not only enhances the quality of our manuscript but also identifies potential avenues for future research expansion in our concluding remarks. We would like to thank you for guiding us to enhance the scope and depth of our work, and for setting out a clear pathway for forthcoming investigations.

Comment 3:

Tables 5–10: Mention the units of CVRMSE and NMAE.

Response to comment:

We appreciate the opportunity to clarify and enhance the readability of our manuscript as suggested. Please be advised that the units for CVRMSE and NMAE are percentages (%) in Tables 5–10. Additionally, we have extended this clarification to Tables 11 to 14, where the units are also in percentages (%). Thank you for your helpful comments that have guided these improvements.

---

## [Decision Letter · Decision Letter 2]

10 Jul 2024

Advancing ensemble learning techniques for residential building electricity consumption forecasting: Insight from explainable artificial intelligence

PONE-D-24-08032R2

Dear Dr. Nam,

We’re pleased to inform you that your manuscript has been judged scientifically suitable for publication and will be formally accepted for publication once it meets all outstanding technical requirements.

Kind regards,

Zeyar Aung

Academic Editor

PLOS ONE

Additional Editor Comments (optional):

Reviewers' comments:

Reviewer's Responses to Questions

**Comments to the Author**

1. If the authors have adequately addressed your comments raised in a previous round of review and you feel that this manuscript is now acceptable for publication, you may indicate that here to bypass the “Comments to the Author” section, enter your conflict of interest statement in the “Confidential to Editor” section, and submit your "Accept" recommendation.

Reviewer #2: All comments have been addressed

2. Is the manuscript technically sound, and do the data support the conclusions?

Reviewer #2: Yes

3. Has the statistical analysis been performed appropriately and rigorously? 

Reviewer #2: Yes

4. Have the authors made all data underlying the findings in their manuscript fully available?

Reviewer #2: Yes

5. Is the manuscript presented in an intelligible fashion and written in standard English?

Reviewer #2: Yes

6. Review Comments to the Author

Reviewer #2: The authors have successfully addressed all my comments and I hereby recommend for the publication of the manuscript.

7. PLOS authors have the option to publish the peer review history of their article (what does this mean?). If published, this will include your full peer review and any attached files.

Reviewer #2: No

---

## [Editor Report · Acceptance letter]

12 Sep 2024

PONE-D-24-08032R2 

PLOS ONE

Dear Dr. Nam, 

I'm pleased to inform you that your manuscript has been deemed suitable for publication in PLOS ONE. Congratulations! Your manuscript is now being handed over to our production team.

Kind regards, 

on behalf of

Dr. Zeyar Aung 

Academic Editor

PLOS ONE